# Reasoning on auto-verifiable, scalable, multi-step synthetic tasks.

## Abstract

Complex reasoning—central to intelligent behavior—demands capabilities beyond pretrained knowledge in large language models (LLMs). Prevailing efforts to improve LLM reasoning often bootstrap from predefined question–answer pairs, selecting high-quality traces to guide self-improvement, which does not scale due to the need for curated problems and solutions. We address this limitation by introducing VAST: Reasoning on Auto-verifiable, Scalable, multi-step synthetic Tasks. VAST enables scalable improvement through structured task that grow in difficulty and allow algorithmic verification of both final answers and intermediate steps, substantially minimizing costly human annotation. We present two complementary methods that leverage VAST: (1) $\nu$GRPO, an online approach that constructs rewards from solver outputs and shows generalization to out-of-distribution tasks; and (2) $\nu$MCTS, a Monte Carlo Tree Search method that derives intermediate rewards from rollout-based solution discovery, thereby enabling self-improvement without human annotation. Together, VAST and these methods provide a practical path to robust, scalable, and verifiable multi-step reasoning in modern LLMs.

## 1 Introduction

**Enabling complex reasoning.** Complex reasoning—the capacity to *systematically* decompose intricate problems, reason step by step [46, 58], and verify intermediate and final solutions—underpins competent intelligent behaviour [16, 50]. While recent Large Language Models (LLMs) display impressive progress in proof discovery [39, 31, 18, 41] and algorithmic planning [49, 21, 3, 14, 33, 42], pretraining alone—despite imbuing models with extensive implicit knowledge—remains insufficient for robust multi-step reasoning [9, 28, 44, 15]. Consequently, post-training strategies that *bootstrap* reasoning as a transferable meta-skill have become central [53, 25, 34, 37, 22].

**Limitations of current supervision.** Bootstrapping methods typically depend on **(question, answer)** pairs or reward models [12, 61, 29, 2] to select high-quality traces, followed by targeted training on those traces [53, 52, 37, 7, 54, 13, 19]. Recent work pushes toward outcome-based *process supervision*, deriving dense signals from final-answer labels [43, 48, 5, 4, 24] and minimizing dependence on manual step annotations [22]. Yet two supervision bottlenecks persist: ▶ **[C1] Readily-available questions**—tasks must be phrased so they naturally evoke a reasoning process; ▶ **[C2] Annotated answers**—verified final answers (or reward models trained on them) are required to identify useful traces. Both assumptions break down in challenging domains: obtaining accurate answers can be prohibitively expensive *[C2]*, and crafting questions that reliably elicit sophisticated reasoning is itself nontrivial *[C1]*. We seek a framework that *learns to reason without predefined questions or human-verified answers*, preserving rigorous feedback with minimal human effort.

**Auto-verifiable, scalable, multi-step synthetic tasks.** We study a family of tasks, called VAST, generated by a program $\mathcal{T}_\alpha$ and controlled by a complexity parameter $\alpha$. A task is *auto-verifiable* if it exposes two fast procedures: (i) a *feasibility checker* that determines for any *partial or complete* state whether constraints are satisfied, enabling the pruning or penalization of invalid actions; and (ii) an *outcome evaluator* that assigns an objective value to any *complete* solution, allowing comparisons among candidates. A task is *scalable* if increasing $\alpha$ monotonically enlarges the search space while keeping the costs of feasibility checking and outcome evaluation low, decoupling difficulty from annotation effort. It is *multi-step* when solving requires many interdependent decisions with long-horizon credit assignment. We call these programmatically instantiated problems *synthetic* because

they can be generated on demand to elicit or refine specific reasoning skills. In many VAST, we can take advantage of *algorithmic solvers* that, under a time or resource budget, searches for a high-quality *completion* from a given partial state by leveraging the checker and evaluator.

**Anchoring ideas.** Consider a quadratic assignment problem where $n$ facilities are placed at $n$ locations to minimize flow–distance cost under one-to-one constraints. A feasibility checker validates permutations for partial or complete assignments and respects fixed placements (constraint checking). An outcome evaluator computes cost by summing flow–distance terms for any complete assignment (solution evaluation). A single parameter $\alpha$ (e.g., number of facilities/locations $n$, flow sparsity, or side constraints) scales combinatorial difficulty while keeping verification cheap and fast (scalable). Each placement prunes choices and shifts pairwise costs through flow interactions; good anchors shape feasibility and cost (multi-step). When tractable, a search routine estimates partial-state value via budgeted heuristic completions using the checker and evaluator—exploited during training. Diverse flow/distance matrices and constraints are programmatically generated at scale (synthetic). Canonical examples of this broader family include optimization tasks [10], puzzle-like problems, and deductive reasoning problems with large combinatorial state–action spaces (see Appendix B), all illustrating auto-verifiable and scalable tasks where locating near-optimal solutions remains challenging.

**Taking advantage of these tasks.** Because feasibility checks give immediate admissibility feedback and outcome evaluators score completed solutions, VAST tasks furnish dense, reliable signals for *policy* learning—without question–answer curation or specialized reward models. We capitalize on these signals with two complementary methods (Sec. 3): *(i) Online RL—νGRPO.* A GRPO-based method that uses the algorithmic solver to estimate values for many *partial* states via *solver-guided* searches for the best attainable completions conditioned on each partial state, then reinforces the most promising trajectories. It

Table 1: Comparison of reasoning paradigms. Our underlying assumption is the existence of $\mathcal{T}_\alpha$, discussed in Section 1.1.

| Method | How to obtain tasks? | How to verify solutions? |
|---|---|---|
| Reasoning with verifiable rewards [22] | $t \sim \mathcal{T}_{\text{fixed}}$, $\mathcal{T}_{\text{fixed}}$ human-collected | Human annotation |
| Reasoning with auto-verifiable and scalable tasks (ours) | $t_\alpha \sim \mathcal{T}_\alpha$, $\mathcal{T}_\alpha$ stochastic task generator based on $\alpha$. | Auto-verified (checker + outcome evaluator) |

also integrates *constraint verification* by penalizing invalid actions. *(ii) Offline RL and search-time inference—νMCTS.* When a strong solver is unavailable or too slow, we run Monte Carlo Tree Search that (a) enforces admissible transitions via the checker, (b) ranks partial solutions using evaluator-backed rollouts, and (c) distills the search experience back into the *policy*.

**Why pursue reasoning when a solver exists?** Even with a capable solver (case (i)), learning a policy is valuable: it amortizes search into a reusable prior that captures task invariances enabling competitive performance. In short, we distill large-scale algorithmic search into portable reasoning abilities within general-purpose models (LLMs), allowing these capabilities to transfer across domains.

**Research questions.** VAST tasks form an ideal test-bed: increasing $\alpha$ scales combinatorial difficulty while keeping verification cheap. We study two main research questions: (1) *Generalization:* Does training on VAST transfer to new instances, distinct task families, and harder regimes (higher $\alpha$)? (2) *Self-improvement:* Can outcome evaluation alone drive iterative policy refinement on self-generated data, without human labels?

⨀ *Conceptual:* We formalize a family of *auto-verifiable, scalable, multi-step synthetic* tasks generated by $\mathcal{T}_\alpha$. Here, *auto-verifiable* means two fast procedures are available: a feasibility checker for partial/complete states and an outcome evaluator for complete solutions. Some tasks additionally admit a budgeted completion solver that we exploit during learning but do not require.
② *Algorithmic:* We introduce *νGRPO* (online) and *νMCTS* (offline and search) that explicitly integrate (a) constraint verification and (b) outcome-based supervision; when tractable, RGRPO further uses solver-guided completions to estimate values for partial states, and both methods distill improvements into the *policy*. ③ *Empirical:* Across diverse VAST tasks (Appendix C), our methods improve reasoning without labeled datasets or specialized reward models; ablations quantify the contribution of constraint verification and the utility of solver-guided completion; transfer experiments test generalization beyond the training generator. ④ *Practical:* When the algorithmic value oracle (search-based solver) is too slow to use in practice, *νMCTS* serves as an effective replacement by coupling cheap constraint checks with targeted exploration and learned comparisons.

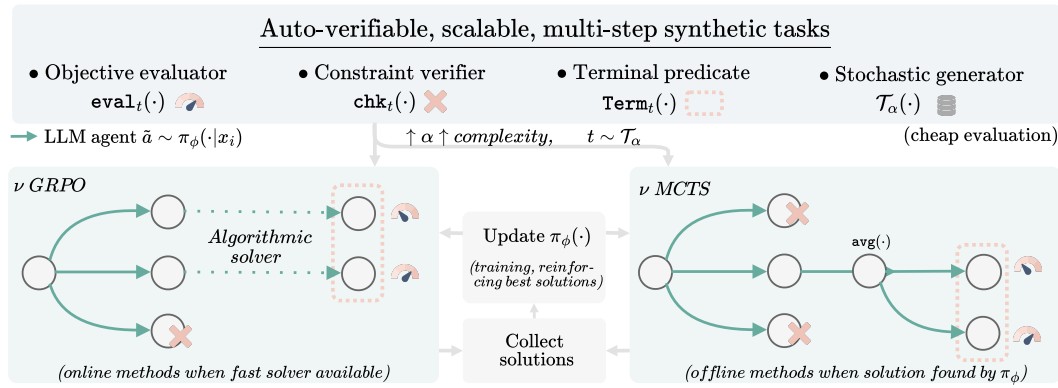

Figure 1: Illustration of VAST framework, depicting structured reasoning on scalable and auto-verifiable tasks via incremental state-action transitions, with intermediate steps algorithmically verified, and outcome solution efficiently evaluated.

## 1.1 AUTO-VERIFIABLE, SCALABLE, MULTI-STEP SYNTHETIC TASKS (VAST)

**Preliminaries.** A *task instance* is a finite- or countable-horizon constrained decision process

$$t = (\mathcal{S}, \mathcal{A}, P, s_0, \tau, \preceq),$$

with state space $\mathcal{S}$, action space $\mathcal{A}$, transition kernel $P(\cdot \mid s, a)$ (deterministic or stochastic), initial state $s_0$, and terminal predicate $\tau : \mathcal{S} \to \{0, 1\}$. We write $\texttt{Term}_t := \{s \in \mathcal{S} : \tau(s) = 1\}$ and $\texttt{Traj}_t(s)$ for all finite trajectories from $s$. The reachability relation $s \preceq s"$ means that $s"$ is reachable from $s$ by some finite sequence of *admissible* actions (Def. 1.1). The encoding length of an object $x$ is $\|x\|$ (bits under a fixed encoding). A task is *multi-step* by default: its minimal feasible horizon $H_t := \inf\{|\rho| : \rho \in \texttt{Traj}_t(s_0), \rho \text{ feasible and ends in } \texttt{Term}_t\}$, satisfies $H_t \geq 1$.

**Definition 1.1** (Task feasibility and admissibility). A feasibility structure for $t$ consists of two predicates, a state checker $\texttt{chk}_t : \mathcal{S} \to \{0, 1\}$ and an action checker $\texttt{chk}_t^{\text{act}} : \mathcal{S} \times \mathcal{A} \to \{0, 1\}$, coupled by the condition

$$\texttt{chk}_t^{\text{act}}(s, a) = 1 \implies \texttt{chk}_t(s) = 1 \quad \text{and} \quad \texttt{chk}_t(s") = 1 \text{ for all } s" \in \text{supp } P(\cdot \mid s, a).$$

Admissible actions are $\mathcal{A}_t(s) := \{a \in \mathcal{A} : \texttt{chk}_t^{\text{act}}(s, a) = 1\}$. A trajectory is *feasible* if it visits only $\texttt{chk}_t$-feasible states and executes only actions in $\mathcal{A}_t(\cdot)$.

**Definition 1.2** (Outcome evaluator). An outcome evaluator is any mapping

$$\texttt{eval}_t : \texttt{Term}_t \longrightarrow \mathbb{R},$$

assumed w.l.o.g. to be maximized (minimization is obtained by negation).

**Definition 1.3** (Auto-verifiable task). A task $t$ is *auto-verifiable* if it is equipped with $(\texttt{chk}_t, \texttt{chk}_t^{\text{act}})$ and $\texttt{eval}_t$ such that: (i) *prefix soundness*—$\texttt{chk}_t(s) = 0$ implies $\texttt{chk}_t(s") = 0$ for all $s" \succeq s$ reachable via admissible transitions; and (ii) *verification efficiency*—there exist polynomials $p_{\text{chk}}, p_{\text{eval}}$ with $\texttt{time}(\texttt{chk}_t(s)) \leq p_{\text{chk}}(\|t\| + \|s\|)$, $\texttt{time}(\texttt{chk}_t^{\text{act}}(s, a)) \leq p_{\text{chk}}(\|t\| + \|s\| + \|a\|)$, and for terminal $s$, $\texttt{time}(\texttt{eval}_t(s)) \leq p_{\text{eval}}(\|t\| + \|s\|)$.

**Value of a partial state.** Let $\Pi_t$ be the policies that choose actions in $\mathcal{A}_t(\cdot)$ and halt only at $\texttt{Term}_t$. The optimal value is

$$V_t^*(s) := \sup_{\pi \in \Pi_t} \mathbb{E}\big[\texttt{eval}_t(S_T) \mid S_0 = s, \pi\big],$$

where $T$ is the (finite) stopping time of first terminal.

**Definition 1.4** (Scalable synthetic generator). A family $\{\mathcal{T}_\alpha\}_{\alpha \in \mathbb{N}}$ of efficient to sample distributions over auto-verifiable tasks is *scalable* if: sampling is polynomial ($\texttt{time}(t \sim \mathcal{T}_\alpha) \leq \text{poly}(\alpha)$ and $\mathbb{E}[\|t\| \mid t \sim \mathcal{T}_\alpha] = \text{poly}(\alpha)$); search-space size grows monotonically with $\alpha$—there exist a measurable size functional $M(t)$, a nondecreasing unbounded $g$, and $\delta(\alpha) \to 0$ such that

$$\mathbb{P}_{t \sim \mathcal{T}_\alpha}\big[M(t) \geq g(\alpha)\big] \geq 1 - \delta(\alpha) \quad (\text{typically } g(\alpha) = \Omega(b^{H_t}) \text{ for some } b > 1);$$

verification remains decoupled from $M(t)$—the time bounds in Def. 1.3 hold with degrees independent of $M(t)$ (hence $\texttt{time}(\texttt{chk}_t), \texttt{time}(\texttt{eval}_t) = \text{poly}(\alpha)$ even when $M(t)$ is combinatorial); and a feasible start exists with high probability (there is at least one feasible terminal reachable from $s_0$ w.p. $1 - \delta_{\text{feas}}(\alpha)$).

Under this scaling, our App.A.1 shows the reasoning budget must grow with task complexity.

**Definition 1.5** (VAST). A family $\{\mathcal{T}_\alpha\}$ defines *VAST* if each $t \sim \mathcal{T}_\alpha$ is auto-verifiable (Def. 1.3) and multi-step ($H_t \geq 1$), and $\{\mathcal{T}_\alpha\}$ is a scalable synthetic generator (Def. 1.4). We refer to $\alpha$ as the *complexity parameter*.

> **Definition 1.6** (Budgeted completion solver (optional oracle)). A (possibly randomized) algorithm $\texttt{Solve}_B$ is a budgeted completion solver if, for any auto-verifiable $t$, feasible partial state $s$ with $\texttt{chk}_t(s) = 1$, and budget $B \in \mathbb{N}$, it returns a multiset $\texttt{Solve}_B(t, s) \subseteq \texttt{Term}_t$ of terminal candidates that are sound (each $s_T$ is terminal and feasible and $\texttt{eval}_t(s_T)$ is exact), budget-monotone (writing $V_B(s) := \max_{s_T \in \texttt{Solve}_B(t,s)} \texttt{eval}_t(s_T)$ with $\max \emptyset = -\infty$, we have $\mathbb{E}[V_B(s)]$ nondecreasing in $B$), and budgeted in cost ($\texttt{time}(\texttt{Solve}_B(t,s)) \leq B \cdot \text{poly}(\|t\| + \|s\|)$ with $\texttt{Solve}_0 = \emptyset$). When available, $\texttt{Solve}_B$ yields an outcome-grounded proxy for $V_t^*(s)$ via best-of-$k$ completions.

**What fits VAST?** See Appendix B.

## 2 Reasoning on VAST Tasks

**From environment dynamics to language reasoning.** VAST tasks are specified at the environment level as constrained decision processes $t = (\mathcal{S}, \mathcal{A}, P, s_0, \tau, \preceq)$ with a feasibility/evaluation relation $\preceq$. Reasoning is performed by an LLM in natural language. To enable this, we expose a language interface that converts states into prompts and parses model outputs back into actions.

**Language interface.** Let $\Sigma$ be the token alphabet and let $\Sigma^*$ denote the set of finite token strings. Let $\mathcal{X} \subseteq \Sigma^*$ denote the set of well-formed *prompt states* (task description, constraints, relevant history), and let $\mathcal{A}" \subseteq \Sigma^*$ denote *reasoning outputs*, i.e., the model's text containing a reasoning trace (enclosed in `<think></think>` tags) and a parsable action payload. The interface comprises a *promptization* map $\psi_\alpha : \Psi_\alpha \times \mathcal{S} \to \mathcal{X}$, where $\Psi_\alpha$ encodes formatting templates parameterized by $\alpha$, that renders a state as a usable prompt, and a deterministic *parser* $\rho : \mathcal{A}" \to \mathcal{A}^{\leq K}$ that extracts up to $K$ structured actions from the *reasoning output*:

$$(t, s) \xrightarrow{\psi_\alpha} x \in \mathcal{X} \xrightarrow{\text{LLM}} \tilde{a} \in \mathcal{A}" \xrightarrow{\rho} a \in \mathcal{A}^{\leq K}.$$

The initial prompt is $x_0 = \psi_\alpha(t, s_0)$. It includes (i) a natural-language task description, (ii) the constraints that must be respected (e.g., admissible actions, budgets), and (iii) any auxiliary instructions (e.g., system guidance). When $K = 1$ the model acts step by step; for $K > 1$ it may emit a short plan in one shot. Importantly, stating constraints in the prompt does not guarantee compliance, which motivates the method in the next section.

**LLM reasoner.** We write $\tilde{a}_i \sim \pi_\phi(\cdot | x_i)$ for the LLM"s text generator conditioned on the prompt-state prefix $x_i$. The prefix summarizes the initial task and the actions already executed, $a_{<i} = \{a_1, \ldots, a_{i-1}\}$.

**State trajectories.** State updates follow the environment dynamics $s_{i+1} \sim P(\cdot | s_i, a_i)$, with current state $s_i$ and action $a_i = \rho(\tilde{a}_i)$. We then reuse $\psi_\alpha$ to construct the next instruction prompt, i.e., $x_{i+1} = \psi_\alpha(t, s_{i+1})$. For simplicity, we write this process as $x_{i+1} \sim \pi_\phi(\cdot | x_i)$. This yields a natural evolution of prompts. The language-level evolution mirrors the environment-level trajectory:

$$x_0 \xrightarrow{\tilde{a}_0} x_1 \xrightarrow{\tilde{a}_1} \cdots \xrightarrow{\tilde{a}_{N-1}} x_N \implies s_0 \xrightarrow{a_0} s_1 \xrightarrow{a_1} \cdots \xrightarrow{a_{N-1}} s_N, \quad (1)$$

where $s_N \in \texttt{Term}_t$ is terminal. Since $\pi_\phi$ may be stochastic and $P$ may be stochastic, many prompt/state trajectories are possible.

### 2.1 What constitutes "better"" reasoning?

Let $V_t^\star(s) := \sup_{\pi \in \Pi_t} \mathbb{E}[\texttt{eval}_t(S_T) \mid S_0 = s, \pi]$ be the optimal value of a partial state (Def. 1.1–1.2).Intuitively, a reasoning act $\tilde{a}$ at prompt state $x_{<i}$ (representing $s_i$) is *better* if it leads, in expectation, to a next environment state with higher downstream value.

**(A) Ideal oracle objective (partial-state value).** If a perfect partial-state oracle $V_t^\star$ were available, the step-optimal reasoning act would satisfy

$$\tilde{a}_i^\star \in \arg\max_{\tilde{a}\in\tilde{\mathcal{A}}} \mathbb{E}_{\substack{\tilde{a}\in\tilde{\mathcal{A}}: \text{ chk}_t^{\text{act}}(s_i,\rho(\tilde{a}))=1 \\ S" \sim P(\cdot \mid s_i,a)}} \Big[ V_t^\star(S") \Big] \approx \arg\max_{\tilde{a}\in\tilde{\mathcal{A}}} \widehat{\mathcal{E}}(t,x_i,\tilde{a}). \tag{2}$$

Here $\widehat{\mathcal{E}}$ is a computable proxy for $V_t^\star$. When a budgeted completion solver $\text{Solve}_B$ exists (Def. 1.6), a natural estimator is $\widehat{\mathcal{E}}_B(t,x_i,\tilde{a}) = \mathbb{E}_{a,S"}\big[V_B(S")\big]$, where $V_B(s) = \max_{s_T \in \text{Solve}_B(t,s)} \text{eval}_t(s_T)$; $V_B$ is sound and budget-monotone and thus a lower bound that tightens with $B$.

**(B) Model-policy objective (no oracle).** Absent any oracle, we prefer reasoning acts whose *rollouts under a reference policy* (e.g., a previous iterate or the current model) yield higher terminal value:

$$\tilde{a}_i^\star \in \arg\max_{\tilde{a}\in\tilde{\mathcal{A}}: \text{ chk}_t^{\text{act}}(s_i,\rho(\tilde{a}))=1} \mathbb{E}[\text{eval}_t(S_T) \mid S_i \sim P(\cdot \mid s_i,\rho(\tilde{a})),\ S_{i+1:T} \text{ roll out under } \pi_\phi]. \tag{3}$$

In practice, Eq. 3 is estimated by a finite number of rollouts with feasibility enforcement and can be used either as a *score* for single-step supervision or as a *preference signal* in pairwise/ranking objectives. Eqs. 2–3 formalize "better reasoning"" at a prompt state: (A) targets true downstream value; (B) substitutes model-based evidence when no oracle exists.[1]

## 2.2 REASONING OBJECTIVE

Our objective is to maximize the per-step reward by selecting effective reasoning traces. Specifically, we sample states $s_i$ induced by prompts $x_i$ according to a specified strategy, and use the policy $\pi$ to determine the next action(s):

$$J(\phi,\alpha) = \mathbb{E}_{\substack{t\sim\mathcal{T}_\alpha,\ s_i \text{ from a sampler} \\ x_i=\psi_t(s_i,\cdot),\ \tilde{a}_i\sim\pi_\phi(\cdot|x_i)}} \Big[ \tilde{r}_t(\tilde{a}_i) \Big]. \tag{4}$$

This reinforcement-learning objective crystallizes the research questions raised by introducing VAST. Along the scalable synthetic task-generator axis $\mathcal{T}_\alpha$, we examine whether training a reasoner $\pi_\phi(\cdot \mid x_i)$ on these tasks yields two benefits: (1) cross-task generalization beyond the training distribution on VAST—specifically, (a) acquiring new reasoning skills, (b) solving harder variants of related problems, and (c) improving performance on strongly out-of-distribution tasks; and (2) sustained—potentially unbounded—self-improvement on VAST. Along the reward objective axis, we study choices for the reward model $\tilde{r}$: (1) for the value of each action $\tilde{a}$, we consider two instantiations based on Eqs. 3 and 2 (an oracle proxy and an approximate value); (2) to ensure feasibility we adopt the check defined in 1.1. Practical implementations of these ideas are presented in the next section.

## 2.3 DISCUSSION

VAST provides a scalable way to *generate* problems that elicit multi-step reasoning and to *assess* solutions through fast feasibility checks and outcome evaluators. In this sense, VAST complements commonly used math/code corpora by supplying dense, reliable signals on tasks whose difficulty is controlled by a single complexity parameter $\alpha$. Because many optimization and logic problems naturally admit constraint checkers and objective evaluators, the family of applicable tasks is broad (see Appendix C). The extent to which reasoning improves depends on both **the task and the model**: low-$\alpha$ instances may saturate quickly for stronger models, whereas higher-$\alpha$ regimes are more likely to reveal and train transferable reasoning skills if the model is capable of is capable of solving some tasks at first instance. On the contrary, even if faced of a complex problem the model could not learn since is not able to extract signal from it, which highlight the importance of VAST tasks.

## 3 METHODS: $\nu$GRPO AND $\nu$MCTS

In this section we instantiate the framework from §2 with practical algorithms that optimize Eq. 4. The core idea is to turn the inexpensive terminal-only signal from the outcome evaluator $\text{eval}_t(\cdot)$ into

---

[1]We recover process-supervised, preference-based, or RL-style updates depending on how $\widehat{\mathcal{E}}$ or the rollout estimate is used.

informative intermediate feedback that rewards good partial decisions and steers subsequent actions. We convert these signals into step-level rewards $\tilde{r}_t$ (Eq. 4) while coupling learning with constraint verification: actions that violate the feasibility checker are discarded or penalized.

We introduce two complementary methods for VAST: $\nu$GRPO (online) and $\nu$MCTS (offline). $\nu$GRPO applies when partial-state values are computable in polynomial time—e.g., via a budgeted completion solver (Def. 1.6); it scores actions by completing from the resulting partial state and reinforces those leading to higher-value attainable completions. When such a solver is unavailable or too slow, $\nu$MCTS performs policy-guided rollouts to terminal states, constructing intermediate values similar to Math-Shepherd [43] and MCTS-style search [48, 5, 4, 24, 55, 56, 11]. In both cases, the feasibility checker constrains exploration and the evaluator supplies outcome-based learning signals.

*Why online and offline RL?* Online methods often achieve strong performance via on-policy updates [17], but they depend on fast, reliable verification. Offline training, by contrast, can accumulate large repositories of verified reasoning traces because verification is inexpensive; these corpora can then be leveraged to improve the policy.

### 3.1 VAST Group Relative Policy Optimization ($\nu$GRPO)

We instantiate an online method that exploits VAST's fast feasibility checks and outcome evaluator by coupling Group Relative Policy Optimization (GRPO) with the algorithmic solver. Intuitively, $\nu$GRPO uses the solver as a value oracle for *partial states*: from each prompt state $x_i$, we sample multiple candidates $\tilde{a}$, score each by the best completion value obtainable under fixed budget from the corresponding next state $x_{i+1}$, and apply a group-relative policy update.

**State sampler.** Let $t \sim T_\alpha$ be a task instance and $s_0$ its initial state. We build a training set $\mathcal{D}$ of intermediate prompt states by (i) invoking the solver once to obtain one high-value complete solution and collecting the partial states along those trajectories.

**Candidate generation & feasibility filtering.** For each $x_i \in \mathcal{D}$, we sample a group of $G$ continuations $\tilde{a} \sim \pi_{\phi_{\text{old}}}(\cdot \mid x_i)$. From each continuation we parse the action(s) $a$ and apply it to obtain the next state $s_{i+1}$. The feasibility checker labels each transition as valid or invalid; invalid transitions receive a fixed penalty $r_{\text{cmin}}$. Valid transitions are scored by the budgeted value oracle: $r_i \leftarrow V_B(x_{i+1})$. Let the set containing the per-candidate rewards. We convert $\mathcal{R}_i$ into advantages via a group-wise rank transform: sort $\mathcal{R}_i$ in descending order, assign ranks $k = 1, \ldots, G$, and map them to evenly spaced targets, which correspond to the advantage used in GRPO:

$$J_{\nu\text{GRPO}} = \mathbb{E}_{\substack{x_i \sim \mathcal{D} \\ \{\tilde{a}_i\}_{i=1}^G \sim \pi_{\phi_{\text{old}}}(\cdot \mid x_i)}} \left[ \frac{1}{G} \sum_{i=1}^G \frac{1}{|o_i|} \sum_{t=1}^{|o_i|} \left( \min\left(r_{i,t}\hat{A}_{i,t}, \text{clip}_\epsilon\left(r_{i,t}\right)\hat{A}_{i,t}\right) - \beta\,\text{KL}\left(\pi_\phi \parallel \pi_{\text{ref}}\right) \right) \right].$$

where $r_{i,t} = \pi_\phi(\tilde{a}_{i,t} \mid \tilde{x}_{i,<t})\,/\,\pi_{\phi_{\text{old}}}(\tilde{a}_{i,t} \mid \tilde{x}_{i,<t})$, and $\text{clip}_\epsilon(r) = \text{clip}(r, 1-\epsilon, 1+\epsilon)$.

### 3.2 VAST Monte Carlo Tree Search ($\nu$MCTS)

**Collecting data.** We begin from the *root* with problem $t$ and an empty solution $x_0$. A complete solution is constructed through the classical MCTS phases—*selection*, *expansion/evaluation*, and *backup*—repeated for a fixed number of rollouts per problem. Each node stores its visit count $N(\cdot)$ and a running value estimate $V(\cdot)$. **(Selection)** We use PUCT to balance exploration and exploitation. At step $i$, select the child that maximizes

$$x_i^\star = \arg\max_{x_i} \left[ \hat{V}(x_i) \;+\; w \cdot \pi(x_i \mid x_{i-1})\sqrt{N(x_{i-1})}/(1 + N(x_i)) \right], \tag{5}$$

where $\pi(x_i \mid x_{i-1})$ is the model"s density for the action (approximated by the exponential of the averaged log-probability of the tokens that form that step in the sampled text), and $w$ scales the exploration bonus. This implements Eq. (3) pragmatically: $V$ approximates expected terminal value, while the exploration term ensures sufficient coverage.

**Expansion & Evaluation (VAST tasks).** At each selected node, we sample $L$" candidate continuations from the LLM, parse them, and *merge duplicates* that map to the same structured action (if $\rho(\tilde{a}_{i,j_1}) = \rho(\tilde{a}_{i,j_2})$, we keep one randomly). For each child: (i) apply $\text{chk}_t^{\text{act}}$ and *discard* invalid

children; (ii) for valid children, we keep the rollout using *selection*, *expansion*, and *evaluation* steps until a terminal score is reached.

**Backpropagation(VAST tasks).** As in standard MCTS, terminal rewards are propagated to update $\hat{V}(\cdot)$ and visit statistics along the path. A key difference in VAST is that *objective values are only revealed at complete rollouts*, and the space of completions can be exponentially large (especially at higher $\alpha$). We therefore retain and aggregate discovered terminal scores so node values can be *ranked* as new solutions arrive. This *rank* is later to use compute the reward actually used in the tree (see Appendix F for details). For *binary objectives* (correct/incorrect), this reduces to the familiar setting used in prior LLM-MCTS work.

**Training**. After search, we *reinforce the best-achieved trajectories* for each problem. Concretely, we apply supervised fine-tuning on the complete solution(s) with the highest $\mathrm{eval}_t(\cdot)$, though other update rules (e.g., rejection sampling, DPO) are also compatible with the framework. This distills search-time discoveries into $\pi_\phi$, enabling stronger search performance at inference [35, 36]

## 4 RELATED WORK

*Scalable generation*: Recent work scales reasoning via procedural generator-verifier tasks in logic and math [6, 47, 45, 60, 39, 40, 1, 30], commonly provide automatic checks, often with binary pass/fail metrics. Newer approaches [38, 20] offer adjustable difficulty and non-binary reward signals, but mainly focused on outcome based evaluation. *Process supervision:* Online step-feedback has leveraged learned process verifiers [23]. Other offline/self-improvement methods [43, 48, 5, 4, 24, 55, 56, 11] adapt outcome supervision from new rollouts yet typically treat final outcomes as binary. Our VAST formulation explicitly separates a feasibility checker (for partial/complete states) from an outcome evaluator that can be graded (e.g., optimization objectives), and our $\nu$GRPO/$\nu$MCTS convert these inexpensive terminal signals into step-level rewards. A complementary line [57, 59] produces self-generated tasks with verification functions, but again uses predominantly 0/1 outcome rewards. Our work provides a general framework and algorithms for creating dense, correctness-preserving step rewards on auto-verifiable tasks, beyond binary-only objectives.

## 5 EXPERIMENTS

We examine whether auto-verifiable, scalable, multi-step synthetic tasks (VAST) support (i) robust generalization and (ii) sustained self-improvement without relying on a fixed solver, and we include targeted ablations of key components. Specifically, we evaluate the generalization benefits of $\nu$GRPO when a solver is available for a subset of tasks; demonstrate that VAST can generate tasks with controllable difficulty; show that, given sufficient compute, MCTS can match solver-guided reasoning and continue to improve; and ablate step-wise search modules.

Prompt examples, data-generation details, and additional results are in the Appendix. For training, we leverage VAST's flexibility to generate level-wise data by varying the complexity parameter $\alpha$; in some cases we use multiple complexity-scaling parameters (see the Appendix for per-task and per-level details). Per level, we sample 1,000 training problems and 250 test problems; different methods ingest this data differently (details in the corresponding sections and the Appendix). We report results for three models: Qwen2.5-3b-Instruct, Llama3.2-3b-Instruct, and Qwen2.5-3b-Instruct. Overall, improvements are uneven: smaller models generalize on simpler tasks but often fail to improve reasoning on harder tasks.

### 5.1 GENERALIZATION ON SYNTHETIC TASKS WITH $\nu$GRPO

We evaluate generalization in three domains: (i) whether training on one optimization task transfers to related optimization tasks; (ii) whether VAST tasks—especially combinations of optimization problems—improve mathematical reasoning; and (iii) whether VAST tasks that leverage synthetic data transfer to 2D spatial reasoning when tasks are presented in text.

We extend the `verl` implementation with ranked, solver-guided partial solutions to define rewards. Training uses 8 rollouts. Per-problem maximum user prompt length varies (Appendix); the maximum response length is 2,048 tokens. From DAPO [51] we adopt upper/lower $\epsilon$ and dynamic batching.

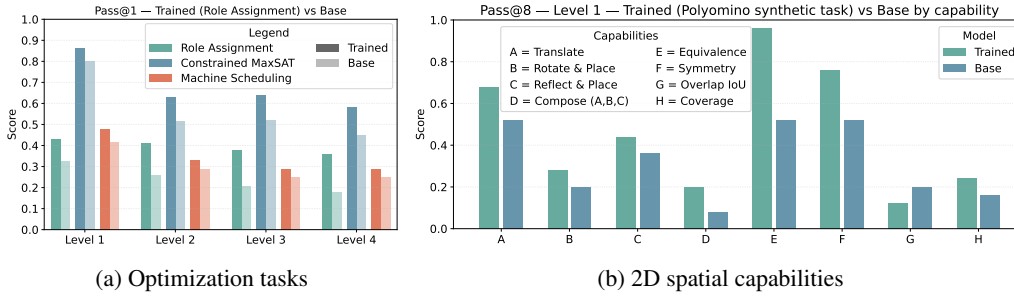

(a) Optimization tasks        (b) 2D spatial capabilities

Figure 2: Model generalization. (a) Training on Role Assignment alone improves performance on unseen optimization tasks (MaxSAT, Scheduling) across all difficulty levels. Light bars show baseline performance; darker colored bars show performance after training. (b) Training on polyomino placement develops eight distinct spatial reasoning capabilities necessary but not explicitly taught.

Before training, a solver constructs optimal paths; for each problem we randomly sample one step along its path. With 1,000 examples per level, the training set contains ~4,000 datapoints. We enforce a curriculum by not shuffling the dataloader (the first block is all "level 1"). At test time we evaluate every step along the path. A prediction is correct if the LLM's action $\tilde{a}$ is a valid step that leads to the best solution. We report pass@$k$ under this metric.

**Optimization tasks.** We assess optimization-to-optimization transfer using Role Assignment (maximize role fitness), Constrained MaxSAT (maximize satisfied clauses), and a machine scheduling problem (choose which task is processed first). Figure 2a shows results when training on Role Assignment and testing on the others. For optimization problems we report pass@1, where gains are largest. The Appendix includes Qwen and Llama 7B results: both improve on pass@1, though gains are smaller for Qwen-7B, suggesting these tasks at this difficulty may be redundant for larger models.

**Mathematical problems.** To test transfer to an out-of-distribution setting, we evaluate whether reasoning acquired from optimization training helps on math benchmarks [26, 27]. Training only on Role Assignment showed no clear gains; combining it with Constrained MaxSAT yields the results in Table 2. Except for Qwen2.5-3B on AIME'24, 3B models improve across datasets, with gains up to 12.5% for Llama 3.2-3B on AMC'23. In the Appendix, gains hold across pass@$k$. We observe no major changes for Qwen2.5-7B.

Table 2: pass@8 (%) on math datasets.

| | Llama 3.2 3B | | Qwen2.5 3B | |
|---|---|---|---|---|
| **Dataset** | **Base** | **Trained** | **Base** | **Trained** |
| **AMC'23** | 45.0% | **57.5%** | 65.0% | **70.0%** |
| **AIME'24** | 10.0% | **16.7%** | 23.3% | 16.7% |
| **AIME'25** | 3.3% | **6.7%** | 6.7% | **16.7%** |

**2D spatial tasks.** Finally, we test whether synthetic data can develop 2D spatial reasoning from text. LLMs are known to struggle with spatial reasoning (e.g., ARC [8]), though recent work shows progress. If we can generate abundant synthetic data, we may mitigate these weaknesses. We therefore build a polyomino-inspired dataset: from many pieces, the model must select, rotate, and translate to maximize the number of *target positions* filled on a 2D grid. Interestingly, Qwen2.5-7B—despite limited gains on earlier tasks—was the only model with sufficient capacity to sample rewardable solutions and bootstrap reinforcement learning here. We test the 2D spatial reasoning capabilities acquired on (1) test set of polyomino

Table 3: Pass@8 (%). L1–L4 denotes curriculum (progressive) training. "L1–L4", "Only L1", and "Only L4" use the same data budget. Pol. = polyomino synthetic task (by level). "2D" aggregates all 2D spatial capabilities.

| | Base | L1–L4 | L1 only | L4 only |
|---|---|---|---|---|
| **Pol. L1** | 4.4% | 10.0% | 32.4% | 4.0% |
| **Pol. L2** | 2.4% | 22.8% | 38.4% | 2.4% |
| **Pol. L3** | 5.3% | 22.5% | 17.1% | 3.6% |
| **Pol. L4** | 1.2% | 5.2% | 2.5% | 0.4% |
| **2D L1** | 32.0% | 46.0% | 28.0% | 29.5% |
| **2D L2** | 24.5% | 34.0% | 31.0% | 27.5% |

and (2) related abilities that the models should have acquired by playing polyomino. We show examples of these task in th Appendix and also additional results to the reported here. Fig. 2b shows level-1 results showcasing using a synthetic task can be helpful to acquire reasoning skill which are related but where not specifically codified, i.e., where capabilities that were necessary to learn to play polyomino. Given the task's difficulty, we study curriculum learning on its test set. Table 3 compares training on only level 4 task (L4), only L1, and a curriculum (L1→L4) under the same data budget. Training only on hard instances fails to acquire basic skills (Pol. L1–L2), making the curriculum the stronger option. Training only on simple instances yields high L1 performance but poor generalization to harder levels and out-of-distribution concepts capabilities.

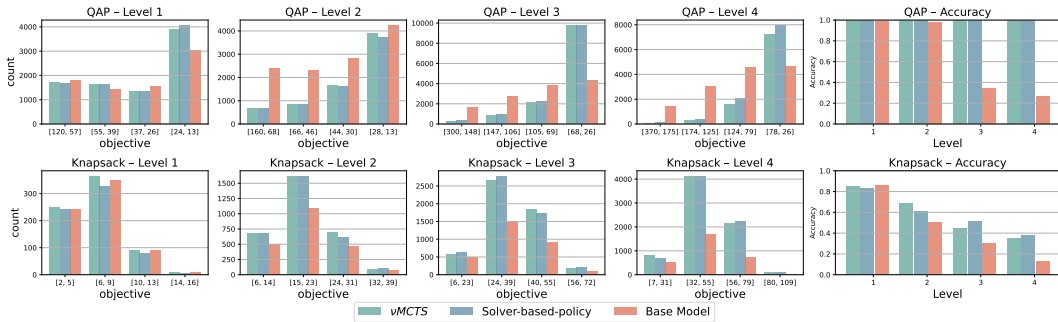

Figure 3: $\nu$MCTS achieves near-solver performance without human labels. Distribution of solution quality (histograms) and accuracy (right) for QAP and Knapsack problems across four difficulty levels. $\nu$MCTS-trained (orange) approach solver-guided performance (green) especially at lower complexity, highlighting self-improvement through verification alone can match algorithmic solvers, eliminating the need for expensive annotations when solvers are not available.

## 5.2 SELF-IMPROVEMENT WITH $\nu$MCTS

For these experiments we use Llama-3.2-Instruct and evaluate on three classical optimization tasks—QAP, Knapsack, and Constrained MaxSAT—reporting full results in the Appendix. Our baselines include basic rejection sampling and SFT on the trace with the highest-utility solution. Training proceeds in stages: we first collect a dataset using MCTS rollouts (branching factor: 20 children per node), then perform SFT. These experiments test the hypothesis that checker+evaluator signals suffice for iterative self-improvement without requiring perfect solver signal.

Because these are optimization problems, multiple valid solutions may exist. Consequently in Figure 3, we report results in two ways: (a) the cumulative number of solutions separated by groups found across 16 rollouts on the test set (first 4 columns), and (b) the percentage of samples that recover the best possible solution (final column). For some tasks (e.g., QAP), with sufficient compute the method maintains performance even as $\alpha$ increases. This is particularly relevant when exact solvers become impractical, as in QAP, which is known to challenge solvers as problem complexity grows. The main observation from this experiment: is that MCTS can match or approximately match of Solver-based-policy, which a reasoning model that is reinforcement with always the solution leading to the best possible outcome.

We ablate the main components of the MCTS. We study the peformance of $\nu$MCTS subtracting the (i) checker (the path is only verified to the very end), and we subtract in addition (ii) the redunday removal of children. Table 4 shows these result for Knapsack observing that in general both components are necessary and can make a big difference in peformance.

Table 4: Ablation MCTS (%)

| Model set | Lvl. 1 | Lvl. 2 | Lvl. 3 | Lvl. 4 |
|---|---|---|---|---|
| $\nu$MCTS | 86.4 % | 50.8 % | 30.4 % | 13.2 % |
| $\nu$MCTS (- checker) | 24.8 % | 0.0 % | 3.2 % | 5.2% |
| $\nu$MCTS (- grouping) | 6.8 % | 1.6 % | 1.2 % | 2.8 % |

## 6 CONCLUSION

We introduce VAST—Reasoning on Auto-verifiable, Scalable, multi-step synthetic Tasks—and two algorithms, $\nu$GRPO and $\nu$MCTS, which turn inexpensive terminal feedback into dense, correctness-preserving step rewards. Across optimization, math, and spatial domains, VAST generalizes to instances and related task families, enables self-improvement without human labels, and remains effective as $\alpha$ grows because verification cost is decoupled from search-space size.

**Limitations & Future work.** VAST lets us vary task difficulty via $\alpha$, a key lever for strengthening reasoning. However, gains are still tied to a pre-defined task distribution. Future work should generate tasks on the fly—e.g., selecting the next most informative instance or composing new VAST tasks from existing ones—to close the remaining automation gap. Recent efforts move in this direction [57]; we view VAST as complementary to them.

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

# A    APPENDIX

## A.1    REASONING AS SOLUTION-DISAMBIGUATING INFORMATION

We formalize "reasoning"" as the amount of information an agent must extract to *disambiguate* the best solution among many feasible completions. This ties the capacity to reason directly to (i) a complexity measure of a task instance—the cardinality of its feasible terminal set—and (ii) the difficulty of distinguishing among many candidate solutions as the task family scales.

**Enumerating terminals.**    For a task instance $t = (\mathcal{S}, \mathcal{A}, P, s_0, \tau, \preceq)$ with feasibility structure $(\texttt{chk}_t, \texttt{chk}_t^{\text{act}})$ and outcome evaluator $\texttt{eval}_t$ (Defs. 1.1–1.2), write the feasible terminal set as

$$\texttt{Term}_t = \{s_T^{(1)}, \ldots, s_T^{(m_t)}\}, \qquad m_t := |\texttt{Term}_t|.$$

Let $I^*(t) \in [m_t]$ be the (tie-broken) index of the unique maximizer $I^*(t) = \arg\max_{i \in [m_t]} \texttt{eval}_t(s_T^{(i)})$;[2] let $V_t^*(s_0) = \texttt{eval}_t(s_T^{(I^*(t))})$.

**Complexity via solution multiplicity.**    We adopt the *solution multiplicity* functional

$$M_{\text{term}}(t) := m_t = |\texttt{Term}_t|$$

as the size measure $M(t)$ in Def. 1.4. For a scalable generator $\{\mathcal{T}_\alpha\}$ (Def. 1.4), there exists a nondecreasing unbounded $g(\alpha)$ and $\delta(\alpha) \to 0$ such that

$$\mathbb{P}_{t \sim \mathcal{T}_\alpha}\big[m_t \geq g(\alpha)\big] \geq 1 - \delta(\alpha). \tag{6}$$

In many VAST, $g(\alpha) = \Omega(b^{H_t})$ with $b > 1$ (Appendix **??**).

**Reasoning transcript and content.**    An agent (policy) $\pi$ interacts with $t$ by taking admissible actions, optionally invoking the checker and evaluator on queried (partial or terminal) states, and halting at some $S_T \in \texttt{Term}_t$. Let $Z^\pi(t)$ be the *transcript* random variable consisting of the entire interaction (trace of visited states, actions, and oracle responses), including the final output $S_T$ as a measurable function of $Z^\pi(t)$. Define the *reasoning content* of $\pi$ on $t$ as the mutual information

$$\mathcal{R}_\pi(t) := I\big(I^*(t); Z^\pi(t)\big) \quad \text{(bits)}, \tag{7}$$

which quantifies how many bits the transcript reveals about the identity of the optimal completion.

For accuracy parameters $\varepsilon \geq 0$ and $\delta \in (0, 1)$, let $\Pi_\alpha(\varepsilon, \delta)$ be the policies satisfying

$$\mathbb{P}_{t \sim \mathcal{T}_\alpha, \pi}\Big[\texttt{eval}_t(S_T) \geq V_t^*(s_0) - \varepsilon\Big] \geq 1 - \delta,$$

and define the *required reasoning budget*

$$\texttt{Req}_{\varepsilon, \delta}(\alpha) := \inf_{\pi \in \Pi_\alpha(\varepsilon, \delta)} \mathbb{E}_{t \sim \mathcal{T}_\alpha}\big[\mathcal{R}_\pi(t)\big]. \tag{8}$$

**Symmetry.**    We will use a mild exchangeability assumption typical of synthetic generators.

**Assumption A.1** (Exchangeability of terminal identities)**.**    Conditional on the set $\texttt{Term}_t$ and their evaluator values $\{\texttt{eval}_t(s_T^{(i)})\}_{i=1}^{m_t}$ (but not on their labels), the generator is invariant to relabelings of the indices $[m_t]$. Consequently, conditional on $m_t$, the maximizer index $I^*(t)$ is uniformly distributed on $[m_t]$, and $H(I^*(t) \mid m_t) = \log_2 m_t$.

**A list-Fano inequality for near-optimality.**    Write the $\varepsilon$-optimal set as $\mathcal{S}_\varepsilon^*(t) = \{i \in [m_t] : \texttt{eval}_t(s_T^{(i)}) \geq V_t^*(s_0) - \varepsilon\}$ with $L_\varepsilon(t) := |\mathcal{S}_\varepsilon^*(t)|$. Success means the agent outputs an index in $\mathcal{S}_\varepsilon^*(t)$. The following generalizes Fano's inequality to this "list-decoding"" notion of correctness.

**Lemma A.2** (Fano with a correct list)**.**    *Fix a task $t$ with $m_t \geq 2$ and let $Z$ be any transcript determining an output $\widehat{I}(Z) \in [m_t]$. Let $P_e := \mathbb{P}[\widehat{I}(Z) \notin \mathcal{S}_\varepsilon^*(t)]$. Then*

$$I\big(I^*(t); Z\big) \geq \log_2 m_t - h_2(P_e) - P_e \log_2(m_t - 1) - (1 - P_e) \log_2 L_\varepsilon(t), \tag{9}$$

*where $h_2(\cdot)$ is the binary entropy function.*

---

[2]Ties have probability zero when $\texttt{eval}_t$ is real-valued with a continuous perturbation; otherwise fix an arbitrary but public tie-breaking rule.

*Proof.* By the chain rule and Assumption A.1, $H(I^*(t)) = \log_2 m_t$. For any (measurable) $Z$, partition on the event of success $E = \{\widehat{I}(Z) \in \mathcal{S}_\varepsilon^*(t)\}$:

$$H\big(I^*(t) \mid Z\big) = H\big(I^*(t), \mathbf{1}_E \mid Z\big) - H\big(\mathbf{1}_E \mid I^*(t), Z\big)$$
$$= H\big(\mathbf{1}_E \mid Z\big) + H\big(I^*(t) \mid Z, E\big)$$
$$\leq h_2(P_e) + P_e \cdot \log_2(m_t - 1) + (1 - P_e) \cdot \log_2 L_\varepsilon(t),$$

since conditioned on $Z$ and failure, $I^*(t)$ can be any of at most $m_t - 1$ indices, and conditioned on $Z$ and success, it must lie within a set of size $L_\varepsilon(t)$. Subtracting from $H(I^*(t))$ yields equation 9. $\square$

**Information lower bounds for reasoning.**    We now state the main consequence for scalable VAST.

**Theorem A.3** (Reasoning required grows with task complexity)**.** *Let $\{\mathcal{T}_\alpha\}$ be a scalable generator (Def. 1.4) with size functional $M_{\text{term}}(t) = |Term_t|$ obeying equation 6. Assume exchangeability (Assumption A.1). Then for any $\varepsilon \geq 0$ and $\delta \in (0, 1)$,*

$$Req_{\varepsilon,\delta}(\alpha) \geq (1 - \delta)\big(1 - \delta(\alpha)\big)\Big(\log_2 g(\alpha) - \Gamma_\varepsilon(\alpha)\Big) - 1, \tag{10}$$

*where $\Gamma_\varepsilon(\alpha)$ is any number satisfying, with probability at least $1 - \delta(\alpha)$ under $t \sim \mathcal{T}_\alpha$,*

$$\log_2 L_\varepsilon(t) \leq \Gamma_\varepsilon(\alpha). \tag{11}$$

*In particular, for* exact *optimality ($\varepsilon = 0$) we may take $\Gamma_0(\alpha) = 0$ when the maximizer is almost surely unique, and*

$$Req_{0,\delta}(\alpha) \geq (1 - \delta)\big(1 - \delta(\alpha)\big)\log_2 g(\alpha) - 1. \tag{12}$$

*Proof.* Fix $\alpha$ and any $\pi \in \Pi_\alpha(\varepsilon, \delta)$. Let $Z = Z^\pi(t)$ be its transcript and $\widehat{I}(Z)$ the index of the terminal solution output by $\pi$. By definition of $\Pi_\alpha(\varepsilon, \delta)$, $P_e := \mathbb{P}[\widehat{I}(Z) \notin \mathcal{S}_\varepsilon^*(t)] \leq \delta$. Apply Lemma A.2 and use $h_2(P_e) \leq 1$ and $\log_2(m_t - 1) \leq \log_2 m_t$:

$$I\big(I^*(t); Z\big) \geq (1 - P_e)\big(\log_2 m_t - \log_2 L_\varepsilon(t)\big) - 1.$$

On the high-probability event $\{m_t \geq g(\alpha),\ \log_2 L_\varepsilon(t) \leq \Gamma_\varepsilon(\alpha)\}$, the RHS is at least $(1 - \delta)(\log_2 g(\alpha) - \Gamma_\varepsilon(\alpha)) - 1$. Taking expectations and using equation 6 gives

$$\mathbb{E}\big[I\big(I^*(t); Z\big)\big] \geq (1 - \delta)\big(1 - \delta(\alpha)\big)\big(\log_2 g(\alpha) - \Gamma_\varepsilon(\alpha)\big) - 1.$$

Since $S_T$ is a function of $Z$, the data processing inequality implies $I(I^*(t); Z) \geq I(I^*(t); S_T)$, so the bound applies a fortiori to $\mathcal{R}_\pi(t) = I(I^*; Z)$. Finally, infimize over $\pi \in \Pi_\alpha(\varepsilon, \delta)$ to obtain equation 10. The exact case equation 12 follows by taking $L_0(t) = 1$ almost surely, hence $\Gamma_0(\alpha) = 0$. $\square$

**Corollary A.4** (Linear growth in horizon under exponential scaling)**.** *Suppose $g(\alpha) = \Omega(b^{H_\alpha})$ for some $b > 1$, where $H_\alpha := \mathbb{E}[H_t \mid t \sim \mathcal{T}_\alpha]$. If there exists a subexponential bound on near-optimal multiplicity, $\Gamma_\varepsilon(\alpha) = o(H_\alpha)$,[3] then for any fixed $\delta \in (0, 1)$,*

$$Req_{\varepsilon,\delta}(\alpha) \geq c \cdot H_\alpha - o(H_\alpha), \qquad c = (1 - \delta)\big(1 - \delta(\alpha)\big)\log_2 b.$$

*Thus the information required to reason grows at least linearly with the effective horizon.*

**Discussion.**    Theorem A.3 grounds the intuition that *more complex tasks demand more reasoning*: as $\alpha$ increases, scalable generators expand the feasible terminal set (larger $m_t$), which raises the entropy of the identity of the best solution; any policy that identifies an (approximately) best completion with high probability must generate transcripts carrying *at least* $\Omega(\log g(\alpha))$ bits of information about that identity. When $g(\alpha)$ grows exponentially with horizon, the required reasoning grows linearly in the horizon, even if an $\varepsilon$-band of near-optimal solutions exists but remains subexponentially large. Notably, the lower bound is agnostic to algorithmic details: exploiting feasibility checkers, evaluator queries, or budgeted completion solvers can only *help* accumulate the necessary information, but cannot beat the information-theoretic requirement imposed by the combinatorics of the solution set.

---

[3]E.g., $\Gamma_\varepsilon(\alpha) = O(\log \alpha)$ when $L_\varepsilon(t)$ is typically polynomial in $\alpha$.

# B    TASK EXAMPLES.

**What fits VAST?** Any multi-step task family with a fast step-checker and a fast final evaluator, whose difficulty scales with a parameter $\alpha$; this spans optimization, logic/constraint puzzles, and simulator-backed domains.

## B.1 MATHEMATICAL OPTIMIZATION TASKS

| Task | State / Actions | D1: Checker (step) | D2: Evaluator (final) | D3: Stop | D4: Scale ($\alpha$) | Typical Solver |
|---|---|---|---|---|---|---|
| **Knapsack** | Subset of items; add/remove item | Total weight $\leq C$ | Total value (max) | All items considered / no improv. | #items, $C$ | DP, ILP, B&B |
| **Bin Packing** | Item→bin assignment; place/open bin | Bin load $\leq$ cap. | #bins (min) | All items placed | #items, capacity | First/Best-Fit, ILP |
| **TSP** | Partial tour; insert/swap city | No repeats; valid edges | Tour length (min) | All cities visited | #cities | 2/3-opt, B&B, Held–Karp |
| **VRP** | Vehicle routes; insert customer | Cap./time-window checks | Total route cost (min) | All customers routed | #cust., vehicles | Clarke–Wright, B&C, meta-heur. |
| **Job Shop** | (job,machine,time) placements | No machine overlap; precedence | Makespan (min) | All ops scheduled | #jobs, machines | ILP, CP, local search |
| **Flow Shop** | Job permutation; shared machine order | Feasible start/finish times | Makespan / flow time (min) | All jobs sequenced | #jobs, machines | Johnson (2-mach), ILP, heur. |
| **Open Shop** | Assign (job, machine, start) | No overlaps on machines | Makespan (min) | All ops assigned | #jobs, machines | CP, ILP, meta-heur. |
| **Cutting Stock** | Patterns; assign demand to patterns | Pattern length $\leq$ stock; demand track | #stocks used (min) | Demands satisfied | Demands, piece types | Column gen., ILP |
| **Facility Location** | Open facilities; assign customers | Capacity per open site | Open+service cost (min) | All customers assigned | Sites, customers | ILP, B&B, heuristics |
| **Max Coverage** | Choose up to $k$ sets | Feasible $|S| \leq k$ | Covered elements (max) | $k$ used / no improv. | #sets, $k$ | Greedy, ILP, local search |
| **Set Cover** | Choose sets covering universe | Coverage tracking | #sets or weight (min) | Universe covered | #sets, elements | Greedy, ILP, B&B |
| **Max Flow / Min Cut** | Edge flows; augment path | Capacity & conservation | $s$–$t$ flow (max) / cut (min) | No augmenting path | Nodes, edges | Edmonds–Karp, Dinic |
| **Assignment** | Agent↔task matching | One-to-one constraint | Total cost (min) | $n$ pairs formed | $n$ | Hungarian, ILP |
| **Quadratic Assignment** | Permutation; swap facilities/locations | Valid permutation | $\sum_{i,j}$ flow·dist (min) | All $n$ placed | $n$, sparsity | B&B, tabu, meta-heur. |

## B.2 LOGIC PUZZLE TASKS

| Task | State / Actions | D1: Checker (step) | D2: Evaluator (final) | D3: Stop | D4: Scale ($\alpha$) | Typical Solver |
|---|---|---|---|---|---|---|
| **Sudoku** | Grid; place/remove digit | Row/col/box constraints | Valid completion (boolean) | Grid full and valid | Grid size, givens | Backtracking + CP |
| **Kakuro** | Runs; fill digits 1–9 | Run sum and no repeats | All runs match sums | All cells filled, all sums met | Grid size, runs | CP, DFS/backtracking |
| **KenKen** | Grid; satisfy cages | Row/col uniqueness; cage op | All cages and rows/cols valid | Full valid grid | Grid size, cage ops | CP, backtracking |
| **Nonogram (Picross)** | Grid; shade/clear cells | Row/col run conformity | All run patterns satisfied | All rows/cols consistent | Grid size, run complexity | Line-solver, CP, SAT |
| **Futoshiki** | Grid with </> | Row/col uniqueness; inequalities | All inequalities satisfied | Full valid grid | Grid size, inequality density | CP, backtracking |
| **Slitherlink** | Edges on lattice; toggle edge | Cell numbers match incident edges; degree $\leq 2$ | Single loop satisfies all clues | Single simple cycle formed | Grid size, clue density | Loop logic, SAT/ILP |
| **Hashiwokakero (Bridges)** | Bridges between islands | No crossings; degree $\leq$ label; $\leq 2$ parallel | All island degrees match; connected | Degrees matched and single component | #islands, labels | Graph heur., CP |
| **Nurikabe** | Shade/clear cells | No 2×2 black; island size $\leq$ label | Island sizes exact; sea connected | All labels satisfied | Grid size, label layout | CP, BFS/DFS logic |
| **Logic Grid (Zebra)** | Attribute matrix; mark yes/no | No-clash and one-of per category; apply clues | All clues satisfied; bijective assignment | All entities fully assigned | #entities, attributes | CP, SAT, tableaux |
| **Mastermind** | Color code; propose guess | Score guess with pegs; consistency with history | Min guesses to exact match | Code guessed or guess limit | Code length, colors | Knuth strategy, search |
| **Hitori** | Cells; black/white decisions | No orthogonal black adjacency; track row/col duplicates | No duplicates; white cells connected | All constraints met | Grid size, digit range | CP, DFS/backtracking |
| **Battleships** | Place fleet; mark water | Row/col ship counts; no adjacency; ship shapes | Fleet placed; counts match | All ships placed and valid | Grid size, fleet mix | CP, ILP/backtracking |
| **Hidato** | Place 1..$N$ consecutively | Each $k$ adjacent to $k\pm1$ | Full 1..$N$ chain | All numbers placed | Grid size, holes | Pathfinding + CP |
| **Tents & Trees** | Place tents near trees | Tent next to a tree; no tent-tent adjacency; row/col counts | Each tree has 1 tent; counts satisfied | All constraints met | Grid size, tree density | CP, logical heuristics |

*Abbrev.:* CP = constraint programming; SAT = satisfiability; ILP = integer linear programming; DFS = depth-first search.

## B.3 SIMULATOR BASED TASKS

| Task | State / Actions | D1: Checker (step) | D2: Evaluator (final) | D3: Stop | D4: Scale ($\alpha$) | Typical Solver |
|------|-----------------|--------------------|-----------------------|----------|---------------------|----------------|
| **Comb. circuit debug/synth.** | Netlist; add/remove gate; rewire; change cell | Width/type compat.; one driver/net; no floating pins; no comb. cycles | Verilog TB sim; score=#tests passed / pass–fail | All tests pass / no improv. | #gates/#nets; depth; #vectors | Verilator / Icarus + static checks |
| **Sequential circuit (bounded)** | Netlist with FFs; edit regs/wires/modules | As left + single-clock; reset well-formed; no multiply-driven state | Simulate $T$ cycles; compare trace vs. spec | Spec satisfied / cycle budget | #FFs+#gates; $T$; #vectors | Verilator (+ STA) |
| **FSM synthesis (DFA/Mealy/-Moore)** | Add states/transitions; label start/accept | Determinism per symbol; well-typed I/O; reachable start | Run labeled traces; score=acc./coverage | All traces satisfied / budget | #states; alphabet; #traces×len | Automata simulator |
| **Sorting network (bounded)** | Append comparators $(i, j)$ on $n$ wires | Indices in range; level constraints; well-formed net | Poly test suite; score=#inputs sorted | Suite fully sorted / budget | #wires; #comparators; suite size | SN simulator / PBT harness |
| **CA target synthesis (Life)** | Toggle seed cells; optional pieces | In-bounds; edit budget | Evolve $T$; score=$-\mathrm{dist}$(target) / exact | Exact match / step budget | Grid size; $T$ | CA engine |
| **Grid-world robot plan** | Append primitives (move/pick/drop) | Parseable; preconds (in-bounds; no collision) | Replay sim; reward=goal − path cost | Goal or horizon | Map size/obstacles; horizon | 2-D grid sim |
| **Compiler pass ordering (IR)** | Sequence optimization passes | Pass applicability; IR verifies/compiles | Run tests; obj.=runtime/size with correct outputs | Tests pass & no further gain / pass limit | IR size; #tests; pass budget | LLVM `opt` + interpreter |
| **Network routing (bounded DES)** | Add/assign routes; flow splits | No simple cycles; link caps. | Simulate $T$; throughput/delay/feas. | All flows delivered or $T$ | #nodes/#edges/#flows; $T$ | Lightweight ns-3–style |

# C  ADDITIONAL DETAILS GENERATION PROCESS.

## C.1  SPATIAL 2D TASKS

**Problem Description**   We generate a family of single-step spatial reasoning tasks on a small $N \times N$ grid ("Canvas"), where cells may be empty ("."), hard obstacles ("#"), target markers ("t"), or previously placed uppercase letters that act as blocking cells. A polyomino shape (triomino, tetromino, or small pentomino) is given together with one or more geometric operations—translation, rotation in $\{0, 90, 180, 270\}°$, and/or reflection (horizontal/vertical/diagonal). An *anchor* $(r, c)$ denotes the top-left position of the shape's tight bounding box on the grid. Legal placement requires all occupied cells to lie within bounds and avoid obstacles or letters.

**Task Catalog**   We export 8 subtasks. Each instance contains a `task_id` and `task_name`; goals are:

1. `translate`: place a given shape by translating from an initial anchor.
2. `rotate_place`: rotate by a prescribed angle and place at a given anchor.
3. `reflect_place`: reflect in a prescribed mode and place at a given anchor.
4. `compose`: rotation + reflection + translation composed, then place.
5. `equivalence`: decide if two shapes are congruent under rotation and optional mirror; if yes, return the rotation and a boolean `reflect`.
6. `symmetry`: report the shape's symmetries (`rot_90`, `rot_180`, `rot_270`, `mirror_h`, `mirror_v`, `mirror_diag`).
7. `overlap_iou`: on an empty visualization grid, place two shapes (with given rotations and anchors) and return intersection cells, intersection count, and union count (IoU).
8. `anchors_covering_point`: at a fixed rotation, list all anchors whose placement covers a specific target cell (sorted).

**Data Generation Process**   For each instance we:

- **Choose a canvas size** $N \times N$ by difficulty (see Table 5).
- **Initialize a grid** with empty cells ".", then sprinkle a small number of obstacles "#" and, at higher levels, decoy letters (from `B`, `C`) that also block placement.
- **Sample a base shape** from predefined libraries: triominoes (3-cell), tetrominoes (4-cell), or a light set of pentominoes (5-cell). Shapes are normalized to tight bounding boxes by removing empty rows/columns.
- **Apply the task-specific transform(s)**: rotations in $\{0, 90, 180, 270\}°$; reflections in `horizontal`, `vertical`, or `diagonal`. Some level-1 cases restrict to fewer rotations/reflections to avoid trivial off-board failures on the smallest canvas.
- **Select anchor(s)** according to the subtask: fixed anchors (given), random legal anchors, the nearest legal anchor to a start point, anchors maximizing target coverage, or the complete set of legal anchors (for enumeration tasks).
- **Ground-truth construction**: each instance includes a one-step oracle answer (`path[0]`) tailored to the subtask—e.g., a final anchor and solution grid, the congruence mapping, a legality verdict with blocked cells, a list of anchors, or the IoU counts. For impossible cases, the oracle sets `impossible = True`.

**Complexity Levels**   Difficulty is controlled primarily by the canvas size, the shape size distribution, and the amount of clutter (obstacles and pre-placed letters): For coverage-style tasks (15–16), the

Table 5: Grid-Spatial complexity levels and typical generation ranges.

| Level | Canvas | `shape_sizes` (cells) | obstacles (#) | decoy_letters (#) | Rotations | Reflections |
|---|---|---|---|---|---|---|
| 1 | 5×5 | 2–3 (mostly triominoes) | 0–1 | 0 | $\{0, 90\}$ or full set | none, horizontal |
| 2 | 6×6 | 3–4 (tetrominoes appear) | 1–2 | 0–1 | $\{0, 90, 180, 270\}$ | horizontal, vertical |
| 3 | 7×7 | 4–5 (tetro/pentominoes) | 1–2 | 1–2 | $\{0, 90, 180, 270\}$ | none, horizontal, vertical, diagonal |

number of targets is typically $4/6/8$ for levels 1/2/3. Tie-breaking follows the implementation: lexicographic by anchor $(r, c)$, and for task 16 by (coverage, rotation, anchor).

**Outputs** Each instance stores the input grid and task parameters under `state`, and a single-step oracle answer in `answer` (identical to `path[0]`). For placement-type tasks we also provide `solution_grid` for visualization. When enabled, images are saved per subtask (`<slug>/images/levelX/<subtask>/`) showing the input and the solution side-by-side with anchors/targets highlighted; task 10 uses a specialized overlay view reporting intersection and IoU.

## C.2   SINGLE MACHINE SCHEDULLING

**Problem Description** We consider the classic single-machine sequencing problem with total weighted tardiness objective: given $n$ jobs with processing times $p_j$, due dates $d_j$, and weights $w_j$, find a permutation minimizing $\sum_j w_j T_j$, where $T_j = \max\{0, C_j - d_j\}$ and $C_j$ is the completion time of job $j$.

**Data Generation Process** For each instance we:

- Sample the number of jobs $n$ and a processing-time range `p_range` from the level specification (Table 6); draw $p_j \sim \text{UniformInt}[\texttt{p\_min}, \texttt{p\_max}]$ and set $P = \sum_j p_j$.

- Draw due dates from the common TF/RDD recipe. With tardiness factor `TF` and relative due-date range `RDD`, we set

$$d_j \sim \text{Uniform}\Big(P\left(1 - \texttt{TF} - \tfrac{\texttt{RDD}}{2}\right), \ P\left(1 - \texttt{TF} + \tfrac{\texttt{RDD}}{2}\right)\Big),$$

then round to integers and clamp $d_j \geq 1$.

- Sample weights via a skewed uniform: $w_j = 1 + \lfloor U^\kappa \cdot (\texttt{Wmax} - 1) \rfloor$, $U \sim \text{Uniform}(0, 1)$, with `Wmax` and exponent $\kappa$ set by level.

- Provide job IDs (A, B, ...) for readability and initialize the schedule prefix `order` to empty.

- **Oracle/ground truth:** given any prefix, we solve the optimal suffix exactly by dynamic programming over (subset,time) to minimize the additional weighted tardiness; the instance includes the optimal full order, the optimal total cost, and a step-by-step `path` consisting of the next jobs to append.

- **Objective exposed to agents:** the per-node objective is the cost of the *scheduled prefix only* (i.e., $\sum w_j T_j$ restricted to the prefix), which is minimized by the search wrapper.

**Complexity Levels** Five difficulty levels control problem size and due-date/weight distributions:

Table 6: Late Fees levels and parameterization.

| Level | $n$ | p_range | TF | RDD | Wmax | $\kappa$ |
|---|---|---|---|---|---|---|
| 1 | 5 | (1, 7) | 0.30 | 0.60 | 3 | 1.0 |
| 2 | 6 | (1, 7) | 0.50 | 0.60 | 5 | 1.0 |
| 3 | 7 | (2, 8) | 0.70 | 0.60 | 10 | 1.2 |
| 4 | 7 | (1, 15) | 0.60 | 0.60 | 6 | 1.0 |

**Outputs** Each instance stores $(p, d, w)$, the empty prefix `order`, and convenience metadata (e.g., $P = \sum p_j$ and the level parameters used). The `answer` equals the instance's optimal total cost; `path` lists the optimal suffix as individual `{job_index}` steps; we additionally include the optimal full order (by indices and by letters) for analysis.

## C.3   POLYOMINO TARGET COVER

**Problem Description** A grid contains *targets* (`t`) and optional *obstacles* (`#`). A small library of labeled polyomino pieces (letters A, B, ...) is provided. In at most $K$ moves, place distinct pieces (no overlap with obstacles or previously placed pieces) to maximize the number of *new* targets covered. Rotations by $0°, 90°, 180°, 270°$ are considered; reflections are not used. The score for a move is $w_t \times$ (targets newly covered), with $w_t = 1$.

**Data Generation Process**     Instances are produced in four stages.

1. **Grid & targets.** Sample a grid of size $H \times W$ with 0–1 random obstacles. Create target *clusters* by stamping small shapes (e.g., $2 \times 2$ square, $T$, $L$, $S/Z$); then *sprinkle* a few singleton targets to increase spatial spread. If no target survives, a small $2 \times 2$ cluster is injected to guarantee at least one target.

2. **Piece pool.** Draw a pool of 3–5 pieces (depending on level) from a library: `domino_only`, `domino_or_square`, or `full` (tetrominoes + selected pentominoes + one $2 \times 3$ rectangle). The sampler optionally forbids certain rotated bounding boxes (e.g., $3 \times 3$, $1 \times 4$), limits others (e.g., at most one $2 \times 3$), and allows low-probability duplicates while respecting per-shape/bbox caps. Pieces are assigned IDs `A`, `B`, `....`

3. **Pre-placed "example" pieces.** To guide the task, 1–2 pieces are pre-placed on the board (not counting against $K$). Their placements are chosen to satisfy:

   - at least a required number of examples cover $\geq 1$ target,
   - a minimum number of *remaining* targets still exists after examples,
   - a subsequent non-zero gain is possible within the move budget $K$.

   On Level 1, example rotations are *frozen to* $0°$; higher levels allow all $0/90/180/270°$.

4. **Oracle label.** The optimal value and plan for the remaining moves are computed with an exact bitboard branch-and-bound that groups placements by target-coverage mask and explores only positive-gain moves. The returned plan is then "decorated" with the grid after each step. The dataset entry stores the resulting `state` (board, pieces, clusters, examples), the optimal additional score `answer`, $K$, and the target count remaining after examples.

Table 7: Polyomino Target Cover: level parameters (abbrev. `spr.`=`sprinkle_singles`, `obs.`=`n_obstacles`).

| Level | grid_size | $K$ | pool_size | piece_set | n_examples | obs. | clusters | spr. |
|---|---|---|---|---|---|---|---|---|
| 1 | (4,4) | 1 | 3 | domino_only | 2 (rot. = $0°$) | (0,0) | 1 (force $2 \times 2$) | (0,1) |
| 2 | (5,5) | 1 | 3 | domino_or_square | 2 | (0,1) | (2,3) | (1,2) |
| 3 | (5,5) | 2 | 4 | full | 2 | (0,1) | (2,3) | (2,3) |
| 4 | (6,6) | 3 | 5 | full | 1 | (0,1) | (3,5) | (2,3) |

**Complexity Levels**    *Notes.* Level 3 forbids rotated bounding boxes $(3,3), (1,4), (4,1), (3,4), (4,3)$ and allows *at most one* $2 \times 3$ (or $3 \times 2$) piece. Level 4 forbids $(3,3), (1,4), (4,1)$ and limits the $2 \times 3$ rectangle to at most one. In all levels $w_t = 1$; reflections are never used.

## C.4   QUADRATIC ASSIGNMENT (CF–EU)

**Problem Description**    We consider a small-$n$ QAP with Manhattan distances on a $g \times g$ grid. Facilities are partitioned into $C$ clusters; pairwise flows are $H$ within cluster and $L$ across clusters. Given a (possibly partial) assignment, the objective is the total cost

$$\sum_{i<j} \text{flow}(i,j) \times \|x_i - x_j\|_1,$$

which the oracle minimizes by completing the partial mapping.

**Data Generation Process**

- Sample $n \in [\texttt{min\_n}, \texttt{max\_n}]$ and assign each facility to one of $C$ clusters.

- Randomly place $\approx \texttt{frac} \times n$ facilities to create a partial assignment; the remaining facilities and free grid locations define the search space.

- Compute the *exact* optimal completion by brute-force enumeration over all bijections of the remaining facilities to free cells; store the optimal cost, full assignment, and a minimal "add" path to complete the current state. For analysis, three random completions are also recorded.

**Complexity Levels**

Table 8: QAP (CF–EU) levels. $C = 2$, $H = 10$, $L = 1$ in all levels; `frac` = 0.25.

| Level | $n$ | grid $g$ | clusters $C$ | preassigned fraction |
|-------|-----|----------|--------------|----------------------|
| 1 | 3 | 4 | 2 | 0.25 |
| 2 | 3 | 5 | 2 | 0.25 |
| 3 | 4 | 5 | 2 | 0.25 |
| 4 | 4 | 6 | 2 | 0.25 |

## C.5 ROLE ASSIGNMENT WITH CONFLICTS

**Problem Description** Assign $R$ roles to $C = R + E$ candidates (one-to-one). Each candidate–role pair has an integer *fit* in $[0, 9]$. Some candidate pairs are mutually costly to select together; penalties are symmetric. The objective (for a full assignment) is

$$\sum_{r=1}^{R} \text{fit}(\text{cand}_r, r) \; - \; \sum_{\{i,j\} \subseteq S} \text{penalty}(i, j),$$

where $S$ is the set of used candidates. Ties are broken in favor of the solution with larger minimum individual fit (*fairness* tie-break).

**Data Generation Process**

1. **Fits.** Sample a $C \times R$ fit matrix in $[0, 9]$; with high probability, inject per-role "standouts" and a few near-ties. Enforce that each role has at least two candidates with fit $\geq 3$.

2. **Conflicts.** Sample pairwise penalties with structural bias (e.g., "chain", "clique-bias", "star" features embedded in the string parameter). Ensure at least one conflict among the greedy unique role cover.

3. **Feasibility & positivity.** Recompute penalties (a few attempts) until the globally optimal score is positive.

4. **Oracle.** Enumerate all $R$-subsets of candidates and all bijections role→candidate consistent with any fixed pairs in the current (possibly partial) state; select the maximizer with the fairness tie-break and emit a minimal completion path.

Table 9: Role Assignment levels ($C = R + E$).

| Level | $R$ | extras $E$ | $C$ | conflict density | penalty range | standout rate | structure |
|-------|-----|-----------|-----|------------------|---------------|---------------|-----------|
| 1 | 3 | 1 | 4 | 0.15 | [1,5] | 0.85 | chain |
| 2 | 4 | 1 | 5 | 0.20 | [2,5] | 0.85 | random |
| 3 | 5 | 1 | 6 | 0.25 | [2,6] | 0.85 | chain overlaps |
| 4 | 6 | 1 | 7 | 0.35 | [3,6] | 0.85 | clique_bias |

**Complexity Levels**

## C.6 WISHED ASSIGNMENTS (CONSTRAINED SELECTION & MATCHING)

**Problem Description** We must choose a subset of tasks and assign each chosen task to a distinct worker, subject to: (i) hard Boolean clauses over tasks (CNF with OR-clauses such as implications $\neg A \vee B$, anti-pairs $\neg A \vee \neg B$, small "at-most-one" groups); (ii) resource budgets (e.g., Money, Time); and (iii) worker eligibility. The primary objective used for learning is to *maximize the total weight of satisfied soft clauses*. For oracle reporting, we use a lexicographic tie-break: maximize soft-weight, then minimize resource usage (Money $\rightarrow$ Time $\rightarrow$ others), then prefer fewer selected tasks.

**Data Generation Process**

1. **Costs & budgets.** For each task, sample per-resource nonnegative integer costs (Money in $[1, 5]$, Time in $[1, 4]$, any additional resource in $[1, 3]$). Budgets are set to $\tau$ times the sum of task costs per resource (rounded, min = 1).

2. **Eligibility.** Either sample each task's eligible workers i.i.d. at rate $p$, ensuring at least one eligible worker per task, or (Level 4) enforce a "role" pattern where tasks alternate roles (e.g., eng/art) and workers alternate availability accordingly.

3. **Hard & soft clauses.** Hard CNF clauses include anti-pairs, implications, and small at-most-one sets; soft clauses mix units, implications, anti-pairs, and "at-least-one of 2/3" with weights in $\{1, 2\}$.

4. **Oracle.** For each $k \leq \min(\#\text{tasks}, \#\text{workers})$, enumerate all $k$-task subsets; check feasibility (hard CNF, budgets, and existence of a perfect matching via Kuhn's algorithm), score by soft-weight and lexicographic tie-break, and keep the best. If no feasible plan appears, budgets are slightly relaxed and (rarely) a conflicting hard clause is dropped as a last resort. The oracle solution (selected set, assignment, scores, resource usage) is stored with the instance.

Table 10: Wished Assignments levels ($\tau = $ budget_tau; densities are per-task).

| Level | tasks | workers | resources | $\tau$ | eligibility | hard dens. | soft dens. | soft weights |
|-------|-------|---------|-----------|--------|-------------|------------|------------|--------------|
| 1 | 4 | 2 | 1 | 0.50 | $p = 0.70$ | 0.30 | 1.20 | {1,2} |
| 2 | 5 | 3 | 1 | 0.55 | $p = 0.65$ | 0.60 | 1.60 | {1,2} |
| 3 | 6 | 3 | 2 | 0.55 | $p = 0.65$ | 0.60 | 1.70 | {1,2} |
| 4 | 7 | 4 | 2 | 0.55 | role: eng/art | 0.70 | 2.00 | {1,2} |

**Complexity Levels**

**Serialization (common to all tasks)** Each instance is stored as a JSON object with a `state` payload that contains the full Markovian description needed for rollouts (board/targets/pieces for Polyomino; partial assignment and cluster metadata for QAP; fits/conflicts for Role Assignment; resources/clauses/eligibility for Wished Assignments). Labels include:

- `answer`: scalar target value used for supervised training (Polyomino: optimal additional score; QAP: optimal cost; Role Assignment: optimal total; Wished Assignments: optimal soft-weight).

- `oracle_path`/`ground_truth`: task-specific optimal plan to complete the current state (minimal "add" sequences), plus tie-break metrics where applicable.

## D PROMPTS (EXAMPLES)

**Constrained MaxSAT (example)**

You are solving a *Wished Assignments* puzzle (Max-SAT + resource assignment) **incrementally**.
**Objective** 1) Maximize the total **weight of satisfied soft clauses** subject to budgets, eligibility, and hard logic.
2) Tie-breaker A: among ties, **minimize resource consumption** lexicographically *(Money, then Time, ...)* [instance-specific]
3) Tie-breaker B: among remaining ties, **minimize the number of selected tasks**.
**Budgets** [instance-specific]
Money=12, Time=11
**Workers** [instance-specific]
W1, W2, W3, W4
**Tasks (costs; eligible workers)** [instance-specific]

- A: (Money=2, Time=1); eligible: {W3} [idx=0]
- B: (Money=2, Time=4); eligible: {W2} [idx=1]
- C: (Money=4, Time=3); eligible: {W1} [idx=2]
- D: (Money=5, Time=4); eligible: {W2, W4} [idx=3]
- E: (Money=1, Time=2); eligible: {W1, W3} [idx=4]
- F: (Money=5, Time=3); eligible: {W2, W4} [idx=5]
- G: (Money=2, Time=3); eligible: {W1, W3} [idx=6]

**(Note: ¬ is negation.)**
**Hard clauses** [instance-specific]

- ¬A ∨ ¬C
- ¬A ∨ G
- ¬C ∨ ¬D
- ¬C ∨ ¬F
- ¬D ∨ ¬F

**Soft clauses (with weight)** [instance-specific]

1. D ∨ ¬F (w=1)
2. C ∨ G (w=1)
3. ¬B (w=1)
4. ¬D ∨ ¬F (w=1)
5. ¬C ∨ F (w=1)
6. ¬A ∨ ¬E (w=2)
7. B ∨ C (w=2)
8. ¬B ∨ C (w=1)
9. ¬B ∨ G (w=2)
10. ¬D ∨ ¬F (w=1)
11. ¬D ∨ E (w=1)
12. G ∨ C (w=2)
13. ¬C ∨ B (w=2)
14. B (w=2)

**Current state** [instance-specific]
Selected tasks (indices) = []
Assigned pairs (task→worker) = []
**Rules**

- Each step, add exactly one task and assign it to one eligible, currently unused worker.
- Never exceed budgets after a step.
- Never violate any hard clause after a step (e.g., respect prerequisites and not-both constraints).

- A worker can be assigned to at most one task; every selected task must be assigned to exactly one eligible worker.
- Think about the marginal effect on soft clauses; sometimes adding a task reduces a soft clause like ¬D.

Briefly explain your reasoning (budget, clause impacts, eligibility), then propose the next action. Immediately after reasoning within `<think>` your reasoning here `</think>`, propose **1** action(s) [instance-specific] with the following format:

```
{"answer":
[
  {
    "task_index": <int>,   # index of the task to select
    "worker_index": <int>  # index of the worker to assign
    (must be eligible and unused)
  }
]
}
```

**Role assignment (example)**

You are solving a **role assignment with conflicts** incrementally.
**Objective** Assign exactly one candidate to each role. Each candidate can take **at most one** role. Your total score is: (sum of chosen fit scores) - (sum of conflict penalties between all selected candidates).
*Tie-break.* If multiple complete assignments tie on total score, prefer the one with the **highest minimum individual fit** across its assigned pairs.
**Current state**

- n_roles = 6, n_candidates = 7 [instance-specific]
- Fits (per candidate → per role) [instance-specific]
  ```
  ["0":  "0":  9, "1":  9, "2":  1, "3":  3, "4":  8, "5":
  3, "1":  "0":  6, "1":  4, "2":  3, "3":  1, "4":  8, "5":
  1, "2":  "0":  1, "1":  0, "2":  1, "3":  7, "4":  9, "5":
  5, "3":  "0":  3, "1":  8, "2":  5, "3":  3, "4":  1, "5":
  3, "4":  "0":  6, "1":  2, "2":  6, "3":  8, "4":  1, "5":
  0, "5":  "0":  8, "1":  7, "2":  4, "3":  1, "4":  8, "5":
  3, "6":  "0":  4, "1":  6, "2":  8, "3":  9, "4":  1, "5":
  1]
  ```
- Conflicts (candidate pairs with penalties) [instance-specific]
  ```
  ["a":  0, "b":  6, "penalty":  3, "a":  0, "b":  5,
  "penalty":  4, "a":  0, "b":  2, "penalty":  6, "a":  0,
  "b":  1, "penalty":  5, "a":  4, "b":  5, "penalty":
  4, "a":  5, "b":  6, "penalty":  3, "a":  1, "b":  5,
  "penalty":  5]
  ```
- Already assigned (role→candidate pairs) [instance-specific]
  ```
  []
  ```

**Rules**

- Propose an assignment for ONE unfilled role only (do not modify previous assignments).
- Do not reuse a candidate already assigned to some role.
- Conflicts are allowed; they simply subtract the listed penalty once per pair of selected candidates.

1) Briefly reason about trade-offs (fit vs. conflicts) and the tie-break.
2) Propose one action following the format below.
Immediately after reasoning within `<think>` your reasoning here `</think>`, propose **1** action(s) [instance-specific] with the following format:

```
{"answer":
[
```

```
  {
    "role": <role_index>,         # integer in
    [0..n_roles-1] not yet assigned
    "candidate": <candidate_idx>  # integer in
    [0..n_candidates-1] not yet used
  }
]
}
```

## Single-machine weighted tardiness scheduling (example)

**Objective**

Minimize total weighted tardiness: $\sum_j w_j \cdot \max(0, C_j - d_j)$.

**Non-preemptive, no-idle rules**

- One machine, one job at a time.
- Once started, a job runs to completion.
- No inserted idle time; schedule is contiguous from time 0.

**Helpful heuristics** (you may use them but still must output a valid action)

- Start from **EDD** (earliest due date) and improve via swaps.
- In a block where both consecutive jobs will be tardy, put the **higher** $w/p$ first.
- Protect high-weight, tight-deadline jobs from being pushed late by long, low-weight jobs.

**State** [instance-specific]

A single machine starts at time 0 and runs **without idle time**. Pick a complete order of all jobs.

*Jobs (with indices):*

| Id | idx | p | d | w |
|----|-----|----|----|---|
| A | 0 | 3 | 9 | 4 |
| B | 1 | 8 | 29 | 4 |
| C | 2 | 11 | 22 | 5 |
| D | 3 | 6 | 14 | 1 |
| E | 4 | 15 | 28 | 2 |
| F | 5 | 7 | 30 | 2 |
| G | 6 | 7 | 19 | 5 |

*Current partial order (prefix):* [] (indices []) [instance-specific]
*Time elapsed so far:* 0 [instance-specific]
*Remaining jobs to place:* ["A","B","C","D","E","F","G"] (indices [0,1,2,3,4,5,6]) [instance-specific]
Immediately after reasoning within <think> your reasoning here </think>, propose **1** action(s) [instance-specific] with the following format:

```
{"answer":
[
  {
    "job_index": <int>  # index of the job you schedule NEXT
  }
]
}
```

## Polyomino Target Cover (example)

You are playing a **Polyomino Target Cover** puzzle **incrementally**.
**Objective & Scoring**

- Maximize the number of targets positions covered by the placed pieces.

**Constraints**

- Choose an **unused** piece, rotate by 0/90/180/270, and place its **top-left** at [row,col].

- All occupied cells must lie inside the board, cannot overlap "#" or existing letters.

**Budget** [instance-specific]

K = 3 total moves; you are at round 0

**General context** You will be given the board **before** and **after** the example piece placement(s), then the inventory of pieces with a note for any example pieces already used.

Local anchor: top-left of the shape bounding box at (0,0) in shape coordinates.

Global anchor: grid cell (r,c) containing that point.

The grid is a 2D list of single-character strings with indexing similar to Python `np.array` lists:

**Board before pieces placement** [instance-specific]

```
initial_grid = [
  [".", ".", ".", ".", ".", "."],
  [".", ".", ".", ".", ".", "."],
  [".", ".", ".", ".", ".", "."],
  [".", ".", ".", ".", ".", "."],
  [".", ".", ".", ".", ".", "."],
  [".", ".", ".", ".", ".", "."]
]
```

**target_locations** [instance-specific]

```
[(0,0), (0,1), (1,0), (1,1), (0,2), (1,2), (2,2),
(0,3), (1,3), (1,4), (2,3)]
```

**Board after example piece placement(s) (current state)** [instance-specific]

```
current_grid = [
  [".", ".", ".", ".", "A", "A"],
  [".", ".", ".", ".", "A", "A"],
  [".", ".", ".", ".", "A", "."],
  [".", ".", ".", ".", ".", "."],
  [".", ".", ".", ".", ".", "."],
  [".", ".", ".", ".", ".", "."]
]
```

**Piece pool** [instance-specific]

```
A (P5), placed at global anchor = (0, 4), rotation = 0,
targets covered = [(1, 4)]:
[
  ["A", "A"],
  ["A", "A"],
  ["A", "."]
]

B (L4):
[
  ["B", "."],
  ["B", "."],
  ["B", "B"]
]

C (Z4):
[
  ["C", "C", "."],
  [".", "C", "C"]
]

D (O):
[
  ["D", "D"],
  ["D", "D"]
]
```

```
E (R6):
[
   ["E", "E", "E"],
   ["E", "E", "E"]
]
```

Immediately after reasoning within <think> your reasoning here </think>, return **exactly ONE** placement with the format:

```
{
   "answer": [
      {
         "piece_id": "<ID>",              # e.g. "B"
         "anchor": [<row>, <col>],        # 0-indexed top-left of the
         transformed shape

         "rotation": <0|90|180|270>,      # degrees, clockwise
         "grid_after": [                  # REQUIRED: board AFTER
         your placement
         (2D list of single-char strings)
            [".", ".", "."],
            ...
         ]
      }
   ]
}
```

**0-1 knapsack (example)**

**Objective**
Add items until no further legal addition can strictly increase the total value.

**Reasoning requirement**
Briefly explain the heuristic guiding your choice (remaining capacity, value/weight ratio, etc.).

**State Representation** [instance-specific]

```
Capacity = 45

Weights = [{"0": 4}, {"1": 18}, {"2": 1}, {"3": 8}, {"4": 12},
{"5": 22}, {"6": 6}, {"7": 22}, {"8": 17}, {"9": 19}, {"10": 4},
{"11": 19}, {"12": 19}, {"13": 16}, {"14": 18}, {"15": 3}]
Values  = [{"0": 1}, {"1": 15}, {"2": 1}, {"3": 10}, {"4": 10},
{"5": 8}, {"6": 5}, {"7": 37}, {"8": 25}, {"9": 27}, {"10": 5},
{"11": 17}, {"12": 21}, {"13": 6}, {"14": 15}, {"15": 1}]

Currently selected item indices = []
Current total weight = 0
Current total value  = 0
```

**Rules**

- Never remove items (0-1 knapsack with only add).

Immediately after reasoning within <think> your reasoning here </think>, propose **1** action(s) [instance-specific] with the following format:

```
{"answer":
[
   {
      "item_index": <index>  # index of item added
   }
]
}
```

**CF-EU Manhattan (example)**

**Grid (Python list-of-lists)** [instance-specific]

```
[
    [".", ".", ".", ".", ".", "."],
    [".", ".", ".", ".", ".", "."],
    [".", "1", ".", ".", ".", "."],
    [".", ".", ".", ".", ".", "."],
    [".", ".", ".", ".", ".", "."],
    [".", ".", ".", ".", ".", "."]
]
```

**Legend**
"." → free cell      "0".."n-1" → facility index

**Assigned facilities (list-of-dict)** [instance-specific]

```
[
    {"facility": 1, "location": [2, 1]}
]
```

**Unassigned facilities** [instance-specific]

$$0, 2, 3$$

**Cluster map** [instance-specific]
```
{"0":  0, "1":  1, "2":  1, "3":  1}
```
**Flow rule** [instance-specific]
H = 10 inside a cluster, L = 1 otherwise.
**Distance metric** [instance-specific]
Manhattan.
**Objective**
Incrementally build a complete assignment that minimises
$\sum_{i<j} \text{flow}(i, j) \times \text{Manhattan}(\text{location}(i), \text{location}(j))$.
Immediately after reasoning within `<think>` your reasoning here `</think>`, propose **1** action(s) [instance-specific] with the following format:

```
{"answer":
[
    {
      "action"  : "add",
      "facility": <unassigned-facility-id>,
      "location": [x, y]  # any currently-free cell
    }
    # continue for each action
]
}
```

# E   POLYOMINO CXAMPLES

(a) Level 1

(b) Level 1

(c) Level 2

(d) Level 2

(e) Level 3

(f) Level 3

(g) Level 4

(h) Level 4

Figure 4: Polyomino target cover examples. This shows the optimal solution found the solver. The first three level have 2 pieces randomly postioned that serves as examples. The last level only one example.

# F   ADDITIONAL DETAILS

## F.1   ADDITIONAL DETAILS OF $\nu$MCTS

In this section, we provide further details on the $\nu$MCTS method. Some content from the main paper is repeated here for clarity and consistency.

Our search algorithm relies on a sequence of rollouts, each consisting of iterative cycles of selection and expansion until a terminal node is reached, after which the obtained reward is backpropagated. A terminal node occurs under one of the following conditions: (1) a final state is found, (2) the maximum search depth is reached, or (3) there are no valid children available (marked as `None`). This approach differs from conventional MCTS, where nodes typically represent either correct or incorrect answers. In optimization tasks, however, there is no inherently incorrect answer, which necessitates adjustments in the reward propagation mechanism, as discussed later.

**Selection.** To effectively balance the exploration of under-visited nodes with exploitation of high-value paths, we adopt the PUCT algorithm [32] during the selection phase. PUCT modifies the estimator $\hat{\mathcal{E}}$ by integrating both the value estimation $\hat{Q}(s_m)$ of the current state $s_m$ and the state visitation frequency $N(\cdot)$. Selection exclusively targets non-terminal nodes, identified by the predicate $\text{Term}(\cdot)$, to ensure terminal nodes contribute rewards only once and thus avoid biasing exploration through repeated propagation of favorable outcomes. Additionally, selection explicitly excludes nodes or children flagged as incorrect by the step verifier $V_{\text{step}}$.

If no valid child nodes exist due to verification failures, subsequent child nodes are marked as `None`. Note that the selection objectives used by UCT and PUCT become deterministic once children are sampled, potentially causing repeated selection of invalid paths. To mitigate this issue, we leverage the exploration component inherent in the PUCT objective by introducing a dedicated counter, $N_{\text{inc}}$, which is incremented and backpropagated whenever this situation arises. The introduction of $N_{\text{inc}}$ ensures that the standard visitation counter $N(\cdot)$ remains unaffected, thus preserving the integrity of the value estimates defined as $\hat{Q}(s_i) = \sum \vec{r}(s_i)/N(s_i)$, where $\vec{r}$ represents the vector of all rewards received for instance $s_i$. The adjusted PUCT objective is then explicitly defined as follows:

$$s_i^* = \arg\max_{s_i}\left[\hat{Q}(s_i) + w\,\pi^*(s_i \mid s_{i-1})\,\sqrt{N(s_{i-1})}\,/\,\left(1 + N(s_i) + N_{\text{inc}}(s_i)\right)\right]. \qquad (13)$$

Here, $\pi_\theta^*(s_i \mid s_{i-1})$ is the exponential of the averaged log-probability of all tokens in the step.

**Expansion.** The expansion procedure is also modified compared to its original formulation. Specifically, since we have access to $V_{\text{step}}$, we can discard invalid solutions. Additionally, due to the inherent structure in the state and action spaces $\mathcal{S}$ and $\mathcal{A}$, equivalent actions can be grouped together.

**Backpropagation.** The final modification applied to the standard Monte Carlo Tree Search (MCTS) algorithm tailored for optimization problems concerns the reward assignment procedure. In optimization scenarios, solutions continuously evolve, and typically the exact range of objective values is unknown in advance, complicating reward normalization. To overcome this, we initially assign a reward of $r = 1$ to the first solution discovered. Subsequently, whenever the model identifies a new solution, all previously backpropagated rewards are recalculated. This recalculation is necessary since new solutions may be superior, inferior, or intermediate relative to the existing solutions. Rewards are consistently maintained within the interval $[m, 1]$.

# G ADDITIONAL RESULTS

## G.1 POLYOMINO SYNTHETIC TASK

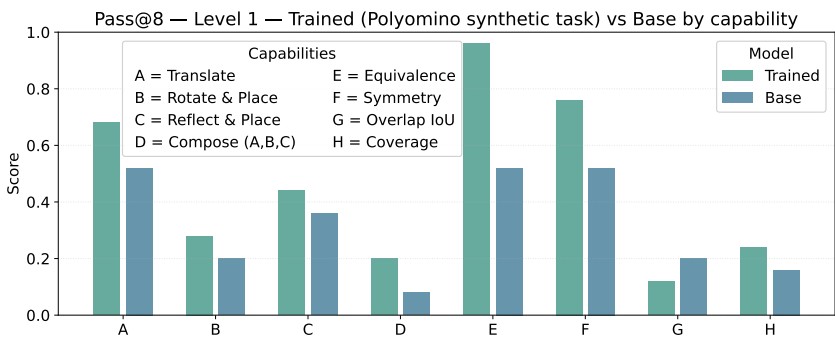

Figure 5: (a)Curriculum Level 1 to 4

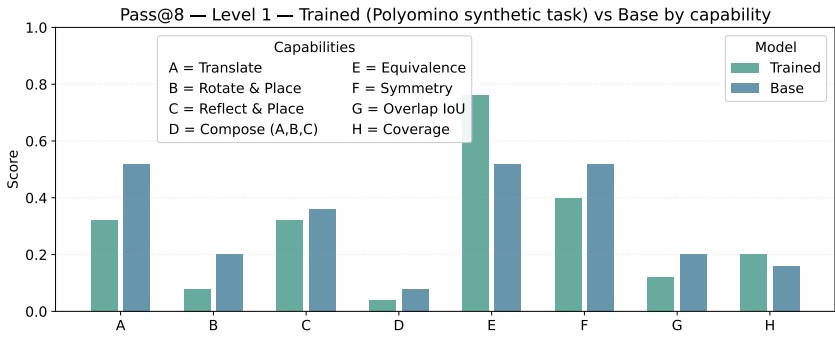

Figure 6: (b) Only Level 1

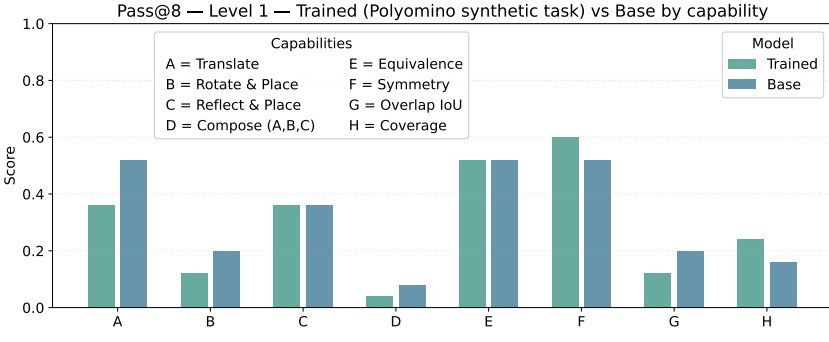

Figure 7: (c) Only Level 4

Figure 8: Test-set pass@8 accuracy, testing spatial reasoning capabilities of level 1 tasks.

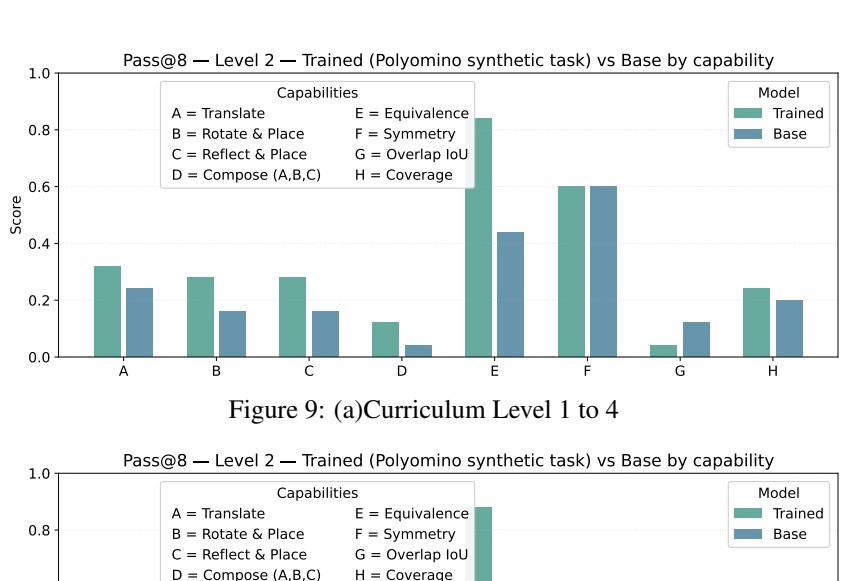

Figure 9: (a)Curriculum Level 1 to 4

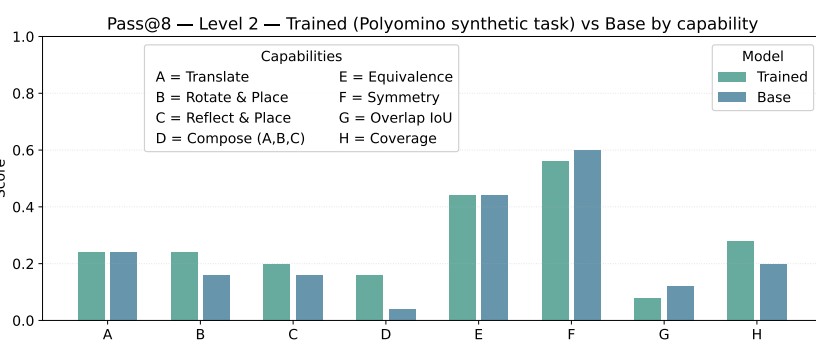

Figure 10: (b) Only Level 1

Figure 11: (c) Only Level 4

Figure 12: Test-set pass@8 accuracy, testing spatial reasoning capabilities of level 2 tasks.

| pass@k | Level 1 | | Level 2 | | Level 3 | | Level 4 | |
|---|---|---|---|---|---|---|---|---|
| | **Base** | **Trained** | **Base** | **Trained** | **Base** | **Trained** | **Base** | **Trained** |
| **Pass@1** | 0.006 | 0.015 | 0.003 | 0.038 | 0.008 | 0.051 | 0.002 | 0.009 |
| **Pass@2** | 0.013 | 0.030 | 0.006 | 0.073 | 0.015 | 0.093 | 0.003 | 0.017 |
| **Pass@3** | 0.018 | 0.044 | 0.009 | 0.105 | 0.022 | 0.126 | 0.004 | 0.024 |
| **Pass@4** | 0.024 | 0.056 | 0.012 | 0.134 | 0.028 | 0.154 | 0.006 | 0.031 |
| **Pass@5** | 0.029 | 0.068 | 0.015 | 0.161 | 0.035 | 0.177 | 0.007 | 0.037 |
| **Pass@6** | 0.034 | 0.079 | 0.018 | 0.185 | 0.041 | 0.196 | 0.009 | 0.042 |
| **Pass@7** | 0.039 | 0.090 | 0.021 | 0.207 | 0.047 | 0.212 | 0.011 | 0.047 |
| **Pass@8** | 0.044 | 0.100 | 0.024 | 0.228 | 0.053 | 0.225 | 0.012 | 0.052 |

Table 11: Polyomino (Base vs Trained) — Progressive training L1-L4

| Capability | Level 1 | | | | Level 2 | | | |
|---|---|---|---|---|---|---|---|---|
| | **Base** | **L1-L4** | **L1 Only** | **L4 Only** | **Base** | **L1-L4** | **L1 Only** | **L4 Only** |
| **Translate** | 0.520 | 0.680 | 0.320 | 0.360 | 0.240 | 0.320 | 0.280 | 0.240 |
| **Rotate & Place** | 0.200 | 0.280 | 0.080 | 0.120 | 0.160 | 0.280 | 0.240 | 0.240 |
| **Reflect & Place** | 0.360 | 0.440 | 0.320 | 0.360 | 0.160 | 0.280 | 0.200 | 0.200 |
| **Compose (A,B,C)** | 0.080 | 0.200 | 0.040 | 0.040 | 0.040 | 0.120 | 0.040 | 0.160 |
| **Equivalence** | 0.520 | 0.960 | 0.760 | 0.520 | 0.440 | 0.840 | 0.880 | 0.440 |
| **Symmetry** | 0.520 | 0.760 | 0.400 | 0.600 | 0.600 | 0.600 | 0.520 | 0.560 |
| **Overlap IoU** | 0.200 | 0.120 | 0.120 | 0.120 | 0.120 | 0.040 | 0.040 | 0.080 |
| **Coverage** | 0.160 | 0.240 | 0.200 | 0.240 | 0.200 | 0.240 | 0.280 | 0.280 |

Table 12: Spatial 2D capabilities — Pass@8 at Level 1 and Level 2.

| Capability | Level 1 | | | | Level 2 | | | |
|---|---|---|---|---|---|---|---|---|
| | **Base** | **L1-L4** | **L1 Only** | **L4 Only** | **Base** | **L1-L4** | **L1 Only** | **L4 Only** |
| **Translate** | 0.140 | 0.205 | 0.070 | 0.090 | 0.090 | 0.125 | 0.085 | 0.095 |
| **Rotate & Place** | 0.050 | 0.085 | 0.010 | 0.035 | 0.075 | 0.070 | 0.080 | 0.085 |
| **Reflect & Place** | 0.190 | 0.190 | 0.145 | 0.165 | 0.055 | 0.075 | 0.045 | 0.045 |
| **Compose (A,B,C)** | 0.020 | 0.035 | 0.005 | 0.005 | 0.015 | 0.035 | 0.010 | 0.030 |
| **Equivalence** | 0.295 | 0.605 | 0.630 | 0.325 | 0.325 | 0.505 | 0.680 | 0.315 |
| **Symmetry** | 0.100 | 0.180 | 0.125 | 0.100 | 0.190 | 0.170 | 0.175 | 0.130 |
| **Overlap IoU** | 0.025 | 0.020 | 0.015 | 0.015 | 0.015 | 0.005 | 0.010 | 0.015 |
| **Coverage** | 0.040 | 0.055 | 0.035 | 0.060 | 0.065 | 0.065 | 0.065 | 0.065 |

Table 13: Spatial 2D capabilities — Pass@1 at Level 1 and Level 2.

## G.2 MATH PROBLEMS

| pass@k | Llama 3.2 3B | | Qwen2.5 3B | | Qwen2.5 7B | |
|---|---|---|---|---|---|---|
| | Base | Trained | Base | Trained | Base | Trained |
| pass@1 | 0.163 | 0.197 | 0.344 | 0.378 | 0.481 | 0.503 |
| pass@2 | 0.252 | 0.296 | 0.456 | 0.487 | 0.605 | 0.609 |
| pass@3 | 0.312 | 0.367 | 0.523 | 0.553 | 0.671 | 0.662 |
| pass@4 | 0.357 | 0.425 | 0.567 | 0.597 | 0.713 | 0.696 |
| pass@5 | 0.390 | 0.473 | 0.598 | 0.629 | 0.741 | 0.721 |
| pass@6 | 0.415 | 0.513 | 0.621 | 0.655 | 0.758 | 0.742 |
| pass@7 | 0.434 | 0.547 | 0.637 | 0.678 | 0.769 | 0.759 |
| pass@8 | 0.450 | 0.575 | 0.650 | 0.700 | 0.775 | 0.775 |

Table 14: AMC"23: pass@k for each model (**Base** vs **Trained**), $k \in \{1, \ldots, 8\}$. Model was trained on joint of dataset Role Assignment and Constrained MaxSAT. The results dataset still follows a sequential order (Level 1 to 4, but each level shuffled).

| pass@k | Llama 3.2 3B | | Qwen2.5 3B | | Qwen2.5 7B | |
|---|---|---|---|---|---|---|
| | Base | Trained | Base | Trained | Base | Trained |
| pass@1 | 0.025 | 0.042 | 0.058 | 0.046 | 0.096 | 0.113 |
| pass@2 | 0.046 | 0.075 | 0.102 | 0.076 | 0.131 | 0.144 |
| pass@3 | 0.064 | 0.101 | 0.136 | 0.098 | 0.150 | 0.165 |
| pass@4 | 0.079 | 0.121 | 0.161 | 0.114 | 0.164 | 0.178 |
| pass@5 | 0.089 | 0.137 | 0.182 | 0.129 | 0.174 | 0.186 |
| pass@6 | 0.096 | 0.149 | 0.200 | 0.142 | 0.183 | 0.192 |
| pass@7 | 0.100 | 0.158 | 0.217 | 0.154 | 0.192 | 0.196 |
| pass@8 | 0.100 | 0.167 | 0.233 | 0.167 | 0.200 | 0.200 |

Table 15: AIME"24: pass@k for each model (**Base** vs **Trained**), $k \in \{1, \ldots, 8\}$. Model was trained on joint of dataset Role Assignment and Constrained MaxSAT. The results dataset still follows a sequential order (Level 1 to 4, but each level shuffled).

| pass@k | Llama 3.2 3B | | Qwen2.5 3B | | Qwen2.5 7B | |
|---|---|---|---|---|---|---|
| | Base | Trained | Base | Trained | Base | Trained |
| pass@1 | 0.004 | 0.013 | 0.021 | 0.037 | 0.079 | 0.087 |
| pass@2 | 0.008 | 0.024 | 0.035 | 0.068 | 0.120 | 0.126 |
| pass@3 | 0.013 | 0.034 | 0.043 | 0.092 | 0.147 | 0.153 |
| pass@4 | 0.017 | 0.043 | 0.050 | 0.112 | 0.169 | 0.171 |
| pass@5 | 0.021 | 0.051 | 0.054 | 0.128 | 0.189 | 0.183 |
| pass@6 | 0.025 | 0.057 | 0.058 | 0.142 | 0.206 | 0.190 |
| pass@7 | 0.029 | 0.062 | 0.062 | 0.154 | 0.221 | 0.196 |
| pass@8 | 0.033 | 0.067 | 0.067 | 0.167 | 0.233 | 0.200 |

Table 16: AIME"25: pass@k for each model (**Base** vs **Trained**), $k \in \{1, \ldots, 8\}$. Model was trained on joint of dataset Role Assignment and Constrained MaxSAT. The results dataset still follows a sequential order (Level 1 to 4, but each level shuffled).

# H OPTIMIZATION PROBLEM

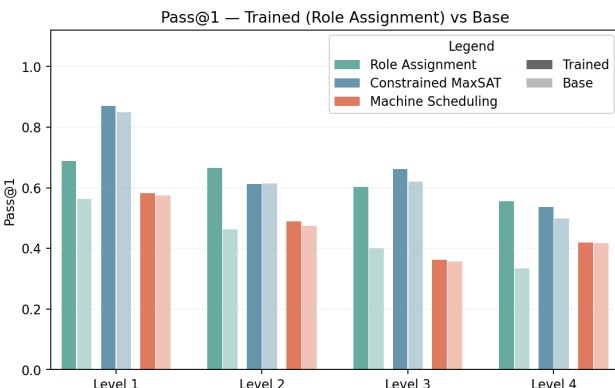

Figure 13: (a) Qwen2.5-3B-Instruct

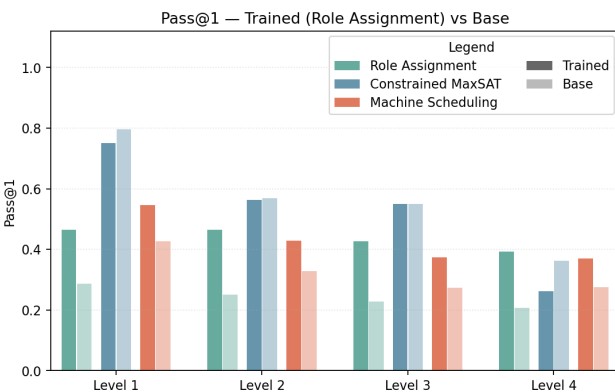

Figure 14: (b) Qwen2.5-7B-Instruct

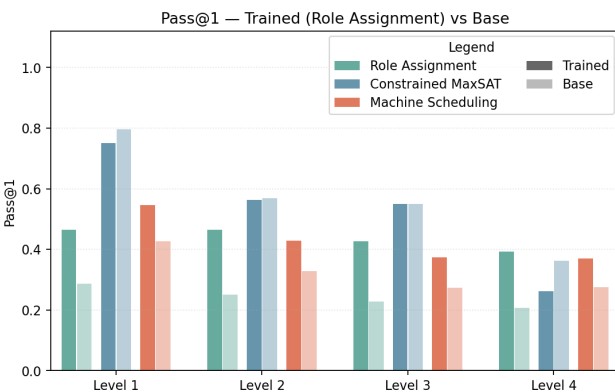

Figure 15: (c) Llama3.2-3B-Instruct

Figure 16: Test-set pass@1 accuracy on Role Assignment, Constrained MaxSAT, and Machine Scheduling, for models fine-tuned on Role Assignment. Each panel reports results for a different instruction-tuned base model.

## H.1  QWEN 2.5 3B

| pass@k | Level 1 | | Level 2 | | Level 3 | | Level 4 | |
|---|---|---|---|---|---|---|---|---|
| | **Base** | **Trained** | **Base** | **Trained** | **Base** | **Trained** | **Base** | **Trained** |
| **Pass@1** | 0.325 | 0.445 | 0.259 | 0.400 | 0.207 | 0.352 | 0.178 | 0.311 |
| **Pass@2** | 0.529 | 0.651 | 0.440 | 0.598 | 0.361 | 0.535 | 0.317 | 0.491 |
| **Pass@3** | 0.661 | 0.765 | 0.569 | 0.713 | 0.479 | 0.646 | 0.427 | 0.607 |
| **Pass@4** | 0.749 | 0.834 | 0.665 | 0.786 | 0.569 | 0.721 | 0.516 | 0.686 |
| **Pass@5** | 0.809 | 0.879 | 0.736 | 0.835 | 0.641 | 0.774 | 0.588 | 0.743 |
| **Pass@6** | 0.852 | 0.910 | 0.791 | 0.871 | 0.699 | 0.813 | 0.649 | 0.786 |
| **Pass@7** | 0.883 | 0.931 | 0.833 | 0.897 | 0.746 | 0.844 | 0.699 | 0.819 |
| **Pass@8** | 0.907 | 0.947 | 0.867 | 0.917 | 0.785 | 0.869 | 0.741 | 0.844 |

Table 17: Role Assignment: Results by Level and Setting (Base/Trained) for pass@k, $k \in \{1, \ldots, 8\}$. Training used Qwen2.5-3B-Instruct.

| pass@k | Level 1 | | Level 2 | | Level 3 | | Level 4 | |
|---|---|---|---|---|---|---|---|---|
| | **Base** | **Trained** | **Base** | **Trained** | **Base** | **Trained** | **Base** | **Trained** |
| **Pass@1** | 0.799 | 0.912 | 0.516 | 0.674 | 0.521 | 0.683 | 0.451 | 0.629 |
| **Pass@2** | 0.908 | 0.965 | 0.720 | 0.825 | 0.713 | 0.824 | 0.649 | 0.780 |
| **Pass@3** | 0.943 | 0.982 | 0.815 | 0.884 | 0.800 | 0.884 | 0.752 | 0.848 |
| **Pass@4** | 0.961 | 0.990 | 0.866 | 0.915 | 0.847 | 0.917 | 0.812 | 0.888 |
| **Pass@5** | 0.972 | 0.994 | 0.898 | 0.934 | 0.876 | 0.938 | 0.851 | 0.914 |
| **Pass@6** | 0.980 | 0.997 | 0.919 | 0.947 | 0.896 | 0.951 | 0.877 | 0.932 |
| **Pass@7** | 0.985 | 0.999 | 0.935 | 0.956 | 0.910 | 0.960 | 0.895 | 0.945 |
| **Pass@8** | 0.989 | 1.000 | 0.948 | 0.962 | 0.922 | 0.966 | 0.908 | 0.956 |

Table 18: Constrained MaxSAT: Results by Level and Setting (Base/Trained) for pass@k, $k \in \{1, \ldots, 8\}$. Training used Qwen2.5-3B-Instruct.

| pass@k | Level 1 | | Level 2 | | Level 3 | | Level 4 | |
|---|---|---|---|---|---|---|---|---|
| | **Base** | **Trained** | **Base** | **Trained** | **Base** | **Trained** | **Base** | **Trained** |
| **Pass@1** | 0.416 | 0.479 | 0.287 | 0.325 | 0.250 | 0.278 | 0.248 | 0.283 |
| **Pass@2** | 0.629 | 0.659 | 0.465 | 0.491 | 0.413 | 0.428 | 0.412 | 0.434 |
| **Pass@3** | 0.749 | 0.746 | 0.582 | 0.586 | 0.525 | 0.518 | 0.526 | 0.525 |
| **Pass@4** | 0.820 | 0.797 | 0.662 | 0.647 | 0.606 | 0.579 | 0.608 | 0.586 |
| **Pass@5** | 0.866 | 0.832 | 0.718 | 0.691 | 0.667 | 0.622 | 0.669 | 0.631 |
| **Pass@6** | 0.897 | 0.858 | 0.760 | 0.724 | 0.714 | 0.657 | 0.716 | 0.666 |
| **Pass@7** | 0.918 | 0.879 | 0.793 | 0.749 | 0.751 | 0.684 | 0.751 | 0.695 |
| **Pass@8** | 0.934 | 0.896 | 0.819 | 0.770 | 0.781 | 0.708 | 0.780 | 0.718 |

Table 19: Machine Scheduling: Results by Level and Setting (Base/Trained) for pass@k, $k \in \{1, \ldots, 8\}$. Training used Qwen2.5-3B-Instruct.

## H.2   QWEN 2.5 7B

| pass@k | Level 1 Base | Level 1 Trained | Level 2 Base | Level 2 Trained | Level 3 Base | Level 3 Trained | Level 4 Base | Level 4 Trained |
|--------|------|---------|------|---------|------|---------|------|---------|
| **Pass@1** | 0.564 | 0.751 | 0.463 | 0.704 | 0.400 | 0.660 | 0.334 | 0.599 |
| **Pass@2** | 0.756 | 0.777 | 0.671 | 0.744 | 0.608 | 0.704 | 0.528 | 0.659 |
| **Pass@3** | 0.845 | 0.786 | 0.780 | 0.761 | 0.727 | 0.723 | 0.651 | 0.686 |
| **Pass@4** | 0.893 | 0.792 | 0.843 | 0.771 | 0.802 | 0.737 | 0.734 | 0.704 |
| **Pass@5** | 0.923 | 0.799 | 0.884 | 0.779 | 0.852 | 0.747 | 0.792 | 0.718 |
| **Pass@6** | 0.943 | 0.804 | 0.911 | 0.786 | 0.885 | 0.755 | 0.835 | 0.729 |
| **Pass@7** | 0.957 | 0.809 | 0.930 | 0.792 | 0.908 | 0.761 | 0.866 | 0.738 |
| **Pass@8** | 0.967 | 0.813 | 0.943 | 0.797 | 0.925 | 0.767 | 0.890 | 0.747 |

Table 20: Role Assignment: Results by Level and Setting (Base/Trained) for pass@k, $k \in \{1, \ldots, 8\}$. Training used Qwen2.5-7B-Instruct.

| pass@k | Level 1 Base | Level 1 Trained | Level 2 Base | Level 2 Trained | Level 3 Base | Level 3 Trained | Level 4 Base | Level 4 Trained |
|--------|------|---------|------|---------|------|---------|------|---------|
| **Pass@1** | 0.849 | 0.961 | 0.614 | 0.808 | 0.620 | 0.813 | 0.498 | 0.793 |
| **Pass@2** | 0.936 | 0.986 | 0.772 | 0.890 | 0.793 | 0.905 | 0.688 | 0.902 |
| **Pass@3** | 0.961 | 0.993 | 0.837 | 0.923 | 0.869 | 0.937 | 0.783 | 0.941 |
| **Pass@4** | 0.972 | 0.996 | 0.872 | 0.942 | 0.911 | 0.953 | 0.837 | 0.962 |
| **Pass@5** | 0.978 | 0.998 | 0.892 | 0.954 | 0.936 | 0.963 | 0.871 | 0.973 |
| **Pass@6** | 0.982 | 0.999 | 0.906 | 0.963 | 0.952 | 0.969 | 0.895 | 0.981 |
| **Pass@7** | 0.984 | 1.000 | 0.916 | 0.970 | 0.962 | 0.973 | 0.912 | 0.986 |
| **Pass@8** | 0.986 | 1.000 | 0.924 | 0.976 | 0.968 | 0.976 | 0.925 | 0.989 |

Table 21: Constrained MaxSAT: Results by Level and Setting (Base/Trained) for pass@k, $k \in \{1, \ldots, 8\}$. Training used Qwen2.5-7B-Instruct.

| pass@k | Level 1 Base | Level 1 Trained | Level 2 Base | Level 2 Trained | Level 3 Base | Level 3 Trained | Level 4 Base | Level 4 Trained |
|--------|------|---------|------|---------|------|---------|------|---------|
| **Pass@1** | 0.575 | 0.585 | 0.474 | 0.510 | 0.357 | 0.352 | 0.418 | 0.427 |
| **Pass@2** | 0.780 | 0.769 | 0.680 | 0.698 | 0.538 | 0.521 | 0.623 | 0.613 |
| **Pass@3** | 0.872 | 0.851 | 0.785 | 0.787 | 0.645 | 0.620 | 0.736 | 0.714 |
| **Pass@4** | 0.919 | 0.895 | 0.845 | 0.837 | 0.714 | 0.684 | 0.805 | 0.775 |
| **Pass@5** | 0.946 | 0.921 | 0.884 | 0.869 | 0.762 | 0.728 | 0.849 | 0.815 |
| **Pass@6** | 0.963 | 0.939 | 0.911 | 0.890 | 0.798 | 0.761 | 0.879 | 0.844 |
| **Pass@7** | 0.974 | 0.952 | 0.930 | 0.906 | 0.824 | 0.786 | 0.900 | 0.865 |
| **Pass@8** | 0.982 | 0.962 | 0.945 | 0.917 | 0.845 | 0.806 | 0.915 | 0.882 |

Table 22: Machine Scheduling: Results by Level and Setting (Base/Trained) for pass@k, $k \in \{1, \ldots, 8\}$. Training used Qwen2.5-7B-Instruct.

## H.3 LLAMA3.2-3B

| pass@k | Level 1 Base | Level 1 Trained | Level 2 Base | Level 2 Trained | Level 3 Base | Level 3 Trained | Level 4 Base | Level 4 Trained |
|---|---|---|---|---|---|---|---|---|
| Pass@1 | 0.287 | 0.435 | 0.251 | 0.514 | 0.228 | 0.494 | 0.207 | 0.441 |
| Pass@2 | 0.482 | 0.468 | 0.424 | 0.553 | 0.392 | 0.533 | 0.359 | 0.492 |
| Pass@3 | 0.616 | 0.484 | 0.548 | 0.572 | 0.513 | 0.554 | 0.474 | 0.520 |
| Pass@4 | 0.711 | 0.495 | 0.639 | 0.585 | 0.605 | 0.567 | 0.563 | 0.538 |
| Pass@5 | 0.778 | 0.502 | 0.706 | 0.595 | 0.675 | 0.576 | 0.634 | 0.552 |
| Pass@6 | 0.825 | 0.508 | 0.757 | 0.603 | 0.730 | 0.583 | 0.690 | 0.562 |
| Pass@7 | 0.860 | 0.513 | 0.796 | 0.609 | 0.773 | 0.589 | 0.736 | 0.571 |
| Pass@8 | 0.884 | 0.517 | 0.825 | 0.614 | 0.806 | 0.594 | 0.774 | 0.577 |

Table 23: Role Assignment: Results by Level and Setting (Base/Trained) for pass@k, $k \in \{1, \dots, 8\}$. Training used Llama-3.2-3B-Instruct.

| pass@k | Level 1 Base | Level 1 Trained | Level 2 Base | Level 2 Trained | Level 3 Base | Level 3 Trained | Level 4 Base | Level 4 Trained |
|---|---|---|---|---|---|---|---|---|
| Pass@1 | 0.796 | 0.946 | 0.568 | 0.755 | 0.550 | 0.737 | 0.362 | 0.710 |
| Pass@2 | 0.912 | 0.972 | 0.766 | 0.825 | 0.738 | 0.809 | 0.540 | 0.784 |
| Pass@3 | 0.946 | 0.982 | 0.849 | 0.858 | 0.819 | 0.841 | 0.640 | 0.816 |
| Pass@4 | 0.963 | 0.986 | 0.893 | 0.881 | 0.863 | 0.860 | 0.702 | 0.837 |
| Pass@5 | 0.973 | 0.989 | 0.918 | 0.898 | 0.889 | 0.874 | 0.745 | 0.853 |
| Pass@6 | 0.980 | 0.991 | 0.935 | 0.911 | 0.906 | 0.885 | 0.776 | 0.866 |
| Pass@7 | 0.985 | 0.992 | 0.947 | 0.922 | 0.918 | 0.895 | 0.802 | 0.877 |
| Pass@8 | 0.989 | 0.993 | 0.955 | 0.931 | 0.927 | 0.902 | 0.824 | 0.886 |

Table 24: Constrained MaxSAT: Results by Level and Setting (Base/Trained) for pass@k, $k \in \{1, \dots, 8\}$. Training used Llama-3.2-3B-Instruct.

| pass@k | Level 1 Base | Level 1 Trained | Level 2 Base | Level 2 Trained | Level 3 Base | Level 3 Trained | Level 4 Base | Level 4 Trained |
|---|---|---|---|---|---|---|---|---|
| Pass@1 | 0.426 | 0.537 | 0.329 | 0.430 | 0.273 | 0.367 | 0.275 | 0.360 |
| Pass@2 | 0.621 | 0.590 | 0.511 | 0.482 | 0.435 | 0.422 | 0.440 | 0.415 |
| Pass@3 | 0.726 | 0.610 | 0.622 | 0.501 | 0.540 | 0.444 | 0.549 | 0.436 |
| Pass@4 | 0.789 | 0.623 | 0.694 | 0.516 | 0.613 | 0.461 | 0.624 | 0.451 |
| Pass@5 | 0.831 | 0.635 | 0.744 | 0.528 | 0.667 | 0.475 | 0.680 | 0.464 |
| Pass@6 | 0.860 | 0.645 | 0.780 | 0.540 | 0.709 | 0.488 | 0.722 | 0.476 |
| Pass@7 | 0.881 | 0.654 | 0.808 | 0.550 | 0.742 | 0.499 | 0.755 | 0.487 |
| Pass@8 | 0.897 | 0.662 | 0.829 | 0.560 | 0.769 | 0.510 | 0.782 | 0.497 |

Table 25: Machine Scheduling: Results by Level and Setting (Base/Trained) for pass@k, $k \in \{1, \dots, 8\}$. Training used Llama-3.2-3B-Instruct.

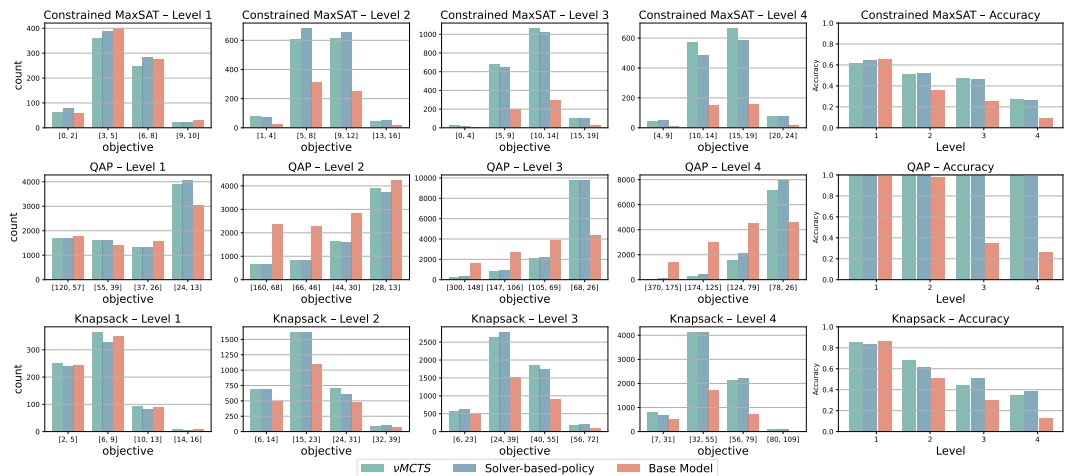

Figure 17: Accuracy and solution-quality histograms for baseline, pure MCTS, and solver-guided MCTS on QAP, 0/1 Knapsack, Constrained MaxSAT tasks as complexity (Level 1 → 4) increases.

## H.4 ADDITIONAL RESULTS MCTS

