# OpenReview forum: "Reasoning on auto-verifiable, scalable, multi-step synthetic tasks"
_ICLR.cc/2026/Conference — ICLR 2026 Conference Desk Rejected Submission_

### Official Review · Reviewer_sVMJ · 2025-10-24

**Soundness:** 2
**Presentation:** 1
**Contribution:** 2
**Rating:** 2
**Confidence:** 2

**Summary:**

This work studies LLM reasoning on VAST, i.e., Auto-verifiable, Scalable, Multi-step Synthetic Tasks. Through vGRPO and vMCTS, the model could generalize to out-of-domain tasks and self-improve.

**Strengths:**

Studies on LLM reasoning from an optimization aspect are less explored.

**Weaknesses:**

1. The writing is hard to follow; providing examples when appropriate would help, e.g., when introducing VAST, what kinds of tasks can be categorized as a VAST. Also, the bullet-style introduction breaks the flow constantly. In Figure 1’s caption, VAST is a framework, but in many other places, VAST refers to auto-verifiable, scalable, multi-step synthetic tasks. Is VAST a method or a task?
2. Also, figure 1 is very distant from the content; it is rarely mentioned, and it is not self-explanatory.
3. While many RLVR tasks need human-annotated labels, those tasks are applicable in assisting humans, such as code generation. Synthesizing data or weak supervision is also possible to reduce such costs. VAST could be a nice test-bed for certain tasks, but the argument that “predefined question–answer pairs does not scale due to the need for curated problems and solutions” is weak.
4.  The math notation is unnecessarily overcomplicated, making the main point even harder to understand. For example, in the definition of an auto-verifiable task, verification efficiency comes from nowhere, and it is also hard to see the necessity of introducing the polynomials p_chk, p_eval. It might be connected to the next “Scalable” part, but it is poorly connected in my opinion.
5. The novelty is limited, which the main framework builds on GRPO and MCTS — both of them are well studied under the context of LLM reasoning.
6. In the intro, the authors mentioned the task is programmed with a parameter alpha controlling the task complexity. However, there is no discussion on how this is controlling the task difficulty, nor how the performance changes w.r.t to this alpha.
7. The authors claim that VAST is scalable, but experiments on scaling such data are missing.
8. Recent work like Reasoning Gym [1] also explores auto-verifiable tasks with controllable task difficulty, and that covers around 100 tasks. What is the main distinction and novelty?

[1] https://arxiv.org/abs/2505.24760

**Questions:**

1. In line 178, after “what fits VAST”,  could briefly mention several classic examples instead of leaving all of them in the appendix.
2. Is there a reason why the related work is placed in Section 4 before the experiment? That also breaks the flow in my opinion.
3. What is the methodological innovation of this work? Both GRPO and MCTS are well explored in the reasoning community, so I wonder what the main novelty is besides using these methods on VAST.
4. How are VAST samples generated?

---

> ### Author Response · Authors · 2025-12-03
>
> Thank you for your helpful feedback and comments. We address each of the points from your review below, but we also included a new version of the manuscript (see above)
>
> ---
>
> ## P1 [Writing]
>
> We agree that the paper was hard to follow, as also noted by other reviewers. In the revised version, we have substantially improved the writing: we have significantly simplified the notation and added illustrative examples to better convey the value of studying reasoning on optimization tasks. We now place more emphasis on the general framework for using online and offline RL in the context of VAST. We have also created a new Figure 1 that more clearly illustrates all components of our paper.
>
> We also agree that there are contexts where we want the LLM to follow an intended question–answer format, which is indeed difficult to scale automatically. We have updated the second paragraph of the introduction to clarify this point.
>
> ---
>
> ## P2 [Novelty]
>
> We agree that methods like GRPO and MCTS are well established. However, $\nu$GRPO and $\nu$MCTS are instantiations of the general techniques we propose for online and offline RL in VAST. To make this clearer, we have rewritten the sections on $\nu$GRPO and $\nu$MCTS as more general Sections 3.1 and 3.2 about online and offline RL.
>
> The novelty lies in the general techniques we develop specifically for VAST tasks and in how we use them to improve step-by-step reasoning in optimization tasks. To our understanding, general online RL methodologies based on a solver to evaluate and distinguish which intermediate steps are better than others have not been used in the context of reasoning.
>
> ---
>
> ## P3 [Controllable parameter $\alpha$ & experiments]
>
> We agree that more discussion of the parameter $\alpha$ is necessary. In the updated version, Section 3.1 now includes an extensive discussion of what fits VAST. In Figure 2 of the same section, we include solver experiments that clearly establish the relationship between $\alpha$ and task difficulty: as $\alpha$ increases, the search space grows and it becomes harder to find the best solution.
>
> We also want to emphasize that scaling experiments were already present in the original version. “Level 1” was intended to denote a specific configuration with lower difficulty than “Level 2”. All experiments in the paper that correspond to VAST tasks (i.e., those generated based on $\alpha$) were run at various difficulty levels. We have now replaced the “Level $i$” notation with $\alpha_i$ to make it clearer that these correspond to different configurations of $\alpha$.
>
> ---
>
> ## P4 [Reasoning Gym]
>
> We agree that Reasoning Gym is closely related to our work and had already discussed it in the related work section of the original version. Compared to our work, Reasoning Gym focuses on proposing a benchmark and does not develop new techniques for improving step-by-step reasoning. In addition, its tasks are generally formulated with binary correctness. By contrast, our focus is on optimization-like reasoning tasks, where many solutions are possible but some are strictly better than others.

---

### Official Review · Reviewer_U7vw · 2025-11-02

**Soundness:** 1
**Presentation:** 2
**Contribution:** 1
**Rating:** 2
**Confidence:** 4

**Summary:**

This paper presents VAST: a set of auto-verifiable reasoning-oriented synthetic tasks with symbolic solvers available to assess correctness of the reasoning and intermediate steps. The authors also show that these tasks can be used with online and offline RL methods like GRPO and MCTS. Lastly, the authors show that this training boosts performance other reasoning tasks of interest that require math abilities or 2D spatial reasoning.

**Strengths:**

- The paper shows generalization from synthetic tasks to mathematical reasoning and 2D spatial reasoning.
- For the most part, the writing is easy to follow and the method is easy to understand

**Weaknesses:**

- The paper does not meaningfully compare to other relevant papers. Apart from the transfer from synthetic planning tasks to math and 2D spatial reasoning, improving the ability of language models at these planning or search oriented tasks is well studied, including past iterations of ICLR, see few examples below and does a poor job of acknowledging these :
     - https://arxiv.org/abs/2404.03683
     - https://arxiv.org/abs/2407.14414
     - https://arxiv.org/abs/2402.14083
     - https://arxiv.org/abs/2402.01817, etc.
- If authors believe that process rewarding is a significant contribution to their method or work (which is possible for other planning tasks too), then they do not engage in prior works suggesting that process rewards are not much more useful than outcome rewards (https://arxiv.org/abs/2402.10963, https://arxiv.org/abs/2403.04642), it also does not ablate training on some other tasks and if they yield more transfer to other reasoning tasks. Furthermore, the authors could use other RL methods like PPO which rely more on process rewards than GRPO.

**Questions:**

In addition to previous points raised, I would personally get rid the paragraph title in the introduction and expand on the related work.

---

> ### Author Response · Authors · 2025-12-03
>
> Thank you for your helpful feedback and comments. We address each of the points from your review below, but we also included a new version of the manuscript (see above)
>
> ---
>
>
> ## P1 [Search and Comparisons]
>
> Our main algorithmic contribution is how to search the large space of optimization tasks. In Section 2.3 we identify three key challenges, and based on these we develop two general techniques to make search more efficient: **(i)** pruning and **(ii)** checking correctness of intermediate steps. These techniques are general and can be applied to any search method.
>
> We have updated the Offline and Online RL sections to clarify that the **main comparisons** in this work are between variants **with and without** these components. This comparison was already present in the original submission and is now summarized in Table 4 of the updated version. Our goal is therefore not to exhaustively compare against all possible search methods, but to evaluate how useful these **search components** are. We clarify this focus in the “Empirical Investigation” section.
>
>
> ---
>
> ## P2 [Benchmarks and Process Rewards]
>
> We agree that comparisons to other benchmarks are important. We view optimization-like reasoning as **complementary** to domains such as code and math. To highlight this, in the updated revision we train on math problems and evaluate optimization reasoning (see Figure 4). This experiment is intended to illustrate that “math reasoning” does not automatically generalize to all domains, and that optimization-style reasoning captures a distinct type of capability.
>
> Regarding process rewards: we believe there is a misunderstanding about how we use them. Standard RL methods like PPO rely on process rewards at the **token-level** MDP (the MDP of language, i.e., actions are the next token to be predicted, and the state is the current generated text). In contrast, in our work we construct intermediate signals in the **problem-level** MDP, i.e., over actions that directly correspond to steps in solving the underlying problem, not rewards assigned to individual tokens as the token-level MDP.
>
> Our methodology is therefore general and applicable to any on-policy RL method; we have made this point clearer in Section 3.1 of the updated version of the paper.

---

### Official Review · Reviewer_RD9h · 2025-11-03

**Soundness:** 2
**Presentation:** 1
**Contribution:** 2
**Rating:** 2
**Confidence:** 4

**Summary:**

This paper introduces VAST, a framework for generating synthetic tasks that are Auto-verifiable, Scalable, and Multi-step，providing tasks with algorithmic verifiers (checkers and evaluators), decoupling solving difficulty from verification cost. The paper proposes two complementary methods, $\nu$GRPO and $\nu$MCTS, to train LLMs using these tasks.

**Strengths:**

VAST introduces a novel optimization perspective by formalizing a task generator where the solving complexity (e.g., exponential search space) is **decoupled** from the low, polynomial verification cost (via $p\_{chk}$ and $p\_{eval}$). This represents a valuable direction for scaling LLM reasoning without expensive human annotation.

**Weaknesses:**

1.	The paper is hard to follow. To make the VAST method more accessible, the authors should consider providing concrete VAST task examples and potentially VAST framework pseudocode in the main body. Furthermore, the discussion of the VAST framework in Figure 1 is vague and misleading, incorrectly emphasizing $\nu$GRPO and $\nu$MCTS over the framework itself.
2.	The experiments fail to demonstrate the specific value of VAST. They show that $\nu$MCTS (self-training) works better than a base model, but not that training on VAST is superior to training on existing synthetic benchmarks using the same algorithms. The value of generating VAST data is not empirically proven.
3.	The paper's core motivation is to address the scalability bottleneck inherent in reasoning training methods that rely on costly, human-curated problem-solution pairs (referring to lines 12-15).However, the paper fails to contextualize this claim by discussing or comparing its $\nu$MCTS self-improvement pipeline (e.g., $\nu$MCTS on VAST) against established unsupervised self-training methods (e.g., PFPO [1], EMPO [2], TTRL [3]) that also do not require human-annotated (question, answer) pairs and thus address the same supervision bottleneck. This lack of comparison prevents a clear assessment of VAST's methodological necessity or superiority.
4.	The central claim of Scalability relies on the theoretical parameter $\alpha$ and the "decoupling" concept (lines 51-53). A major deficiency is the paper's failure to explicitly define the relationship between this theoretical scaling parameter $\alpha$ and the discrete experimental difficulty settings, denoted as "Level 1-4," in the main text.

[1] Jiao F, Guo G, Zhang X, et al. Preference optimization for reasoning with pseudo feedback[J]. arXiv preprint arXiv:2411.16345, 2024.

[2] Zhang Q, Wu H, Zhang C, et al. Right question is already half the answer: Fully unsupervised llm reasoning incentivization[J]. arXiv preprint arXiv:2504.05812, 2025.

[3] Zuo Y, Zhang K, Sheng L, et al. Ttrl: Test-time reinforcement learning[J]. arXiv preprint arXiv:2504.16084, 2025.

**Questions:**

Please see the weakness.

---

> ### Author Response · Authors · 2025-12-03
>
> Thank you for your helpful feedback and comments. We address each of the points from your review below, but we also included a new version of the manuscript (see above)
>
> ---
>
>
> ## P1 [Hard to follow]
>
> We agree that the original version was hard to follow, as also noted by other reviewers. In the revised manuscript, we substantially rewrote the paper: we simplified the notation, added more concrete and illustrative examples of optimization tasks, and clarified the value of reasoning on such tasks. We now more clearly highlight the general VAST framework and how Online and Offline RL are used within it. We also replaced Figure 1 with a new version that explicitly illustrates all key components of the framework rather than emphasizing specific algorithms.
>
> ## P2 [Value of VAST]
>
> Our intention in the original submission was to demonstrate the value of VAST by showing that optimization tasks can be used to improve optimization-like reasoning (Figure 2(a) of the initial version), that multiple skills are learned in the process (Figure 2(b) of the initial version), and that these skills generalize to out-of-distribution tasks. However, we acknowledge that the writing did not make this sufficiently clear, as you pointed out.
>
> In the revised manuscript, we sharpen the presentation of VAST’s value in three ways:
> 1. **Conceptual motivation:** We more explicitly argue that optimization-like reasoning is valuable in its own right, as it requires searching among many candidate solutions—a core human reasoning ability—and that VAST tasks capture a diverse range of such scenarios.
> 2. **Framework clarity:** We improve the explanation of how VAST tasks are constructed and how they differ from standard synthetic benchmarks, emphasizing their auto-verifiable and optimization-oriented nature.
> 3. **Empirical evidence:** In Figure 4 (of the updated version), we include models trained on math-focused baselines, which perform poorly when evaluated on optimization tasks. This directly supports the claim that training on VAST-style optimization tasks is necessary to acquire the targeted optimization-like reasoning capabilities.
>
> ## P3 [Search comparison]
>
> The main algorithmic challenge we address is how to search efficiently over a large space of optimization tasks. In the new Section 2.3 we identify three concrete challenges and, based on these, introduce general techniques such as pruning and step-wise correctness checking to make search more efficient. These techniques are search-agnostic and can, in principle, be paired with different search algorithms.
>
> In the revision, we clarify in the Offline and Online RL sections that our primary goal is to evaluate the benefit of these VAST-specific components (e.g., pruning and checking) rather than to exhaustively compare all possible search methods. Accordingly, the key comparisons focus on ablations with and without these components. This comparison was already present in the original version and is now clearly summarized in Table 4 in the updated manuscript. We explicitly state this focus in the “Empirical Investigation” section.
>
> ## P4 [Scalability]
>
> The relationship between the search space and the complexity parameter is formalized in our definition of scalability in the paper. The exact functional form of this relationship is task-generator–dependent. For this reason, we deliberately keep the definition flexible rather than hard-coding a specific structure, which would limit its applicability to different generators.
>
> We agree, however, that this connection was not easy to understand in the original version. To address this, we have added concrete illustrative examples, in particular in Figure 2 of the updated version. Figure 2 shows classical optimization problems and the reasoning capabilities required to solve them. As the complexity parameter \(\alpha\) increases, both the search space \(M(t)\) (blue line) and the number of unique solutions (teal line) grow. Consequently, the **best solution** (orange line) becomes harder to find, requiring more sophisticated reasoning strategies or heuristics if these tasks are to be solved by a reasoning model.

---

### Official Review · Reviewer_nHKh · 2025-11-10

**Soundness:** 1
**Presentation:** 1
**Contribution:** 1
**Rating:** 2
**Confidence:** 3

**Summary:**

The authors propose VAST, a framework for reasoning on auto-verifiable, scalable, multi-step synthetic tasks. Using this framework, they programmatically generate synthetic tasks that consist of a feasibility checker and an outcome evaluator. The authors then train on these tasks using either an online method -- specifically a GRPO variant or an offline algorithm that does rejection fine-tuning on searched trajectories with MCTS. Most of the experiments are limited to these synthetic tasks with some attempts to transfer to real math benchmarks that lead to rather weak results.

**Strengths:**

* VAST is a framework for synthetic reasoning data generation and training that aims to mitigate the reliance on human-annotated reasoning benchmarks.

**Weaknesses:**

* The experimental section of the paper is extremely weak. The only noteworthy experiments seem to be those in Table 2 in which the authors test their model for generalization to real math tasks. Even then, the presentation of these experiments and the results themselves are quite weak. First, Qwen-2.5-7B shows a 7% drop in performance. Second, it is unclear why this main table reports pass@8 instead of pass@1 and while there is a pointer to appendix, I am not sure which table to look at and what conclusions to draw from it (more on the readability of the paper below). Third, AIME benchmarks are rather small and the authors should clearly specify how many seeds they used. Fourth and most importantly, it does not tell much about the usefulness of the framework -- experiments are with older qwen-2.5 models on only 3 benchmarks, only pertaining to math.

* I had a hard time reading the paper. Things that in my opinion should be central to understanding the paper like describing the synthetic tasks are all moved to appendix. The main text does not even talk about what a typical task looks like. The main paper also lacks sufficient details of the experimental setup and how this work compares to past works on synthetic data. The related work is a small paragraph that does not comment on the contributions of this work beyond what is already present. Instead most of the paper is spent on formalizing notations (some being quite obvious). This is distracting, especially given the weak nature of the experiments. I sincerely hope the authors improve the readability of the paper.

**Questions:**

Please refer to my weakness above.

---

> ### Author Response · Authors · 2025-12-03
>
> Thank you for your helpful feedback and comments. We address each of the points from your review below, but we also included a new version of the manuscript (see above)
>
> ---
> ## P1 [Relevance of Experiments]
>
> We repectfully disagree with the reviewer that the only noteworthy experiment is the generalization to math in Table 2. In fact, we see that experiment as “nice to have” but not central to the main message of the paper.
>
> Our primary contribution is the conceptualization that optimization-like tasks are easy to generate and to verify at scale. We view optimization-like reasoning as complementary to domains such as code and math. To highlight this, we study optimization reasoning by training on math problems (see Figure 4 in the updated version of the paper). This showcases that “math reasoning” is not automatically generalizable to any domain, and emphasizes the distinct value of optimization-style reasoning.
>
> More broadly, we believe the key contribution lies in formalizing the conditions under which VAST tasks can be generated and verified. Under these conditions, data can be generated in essentially unlimited quantities while retaining efficient automatic verification.
>
> ---
>
> ## P2 [Readability and Presentation]
>
> We agree with the reviewer (and with other reviewers) that the paper is currently hard to follow, overly focused on notation, and does not give enough intuitive space to the core ideas and experiments. We take responsibility for this, and we acknowledge that this may have contributed to the impression that Table 2 contains the only interesting results.
>
> In the revised version of the paper, we are restructuring the presentation to reduce unnecessary formalism, bring concrete examples and task descriptions into the main text, and clarify the experimental setup and relation to prior work. We expect these changes to significantly improve the readability and better convey the overall contribution.

---

### Author Response · Authors · 2025-12-03

We thank the reviewers for their comments and feedback.

We have addressed these concerns and uploaded a thoroughly revised version of the paper at the following link: [here](https://limewire.com/d/OlucC#tEvhkkFX0q). We believe many of the issues raised stemmed from unclear writing, which led to misunderstandings about our goals and contributions. In summary:

- We updated the introduction, simplified the notation in Section 2, and added concrete examples. These changes clarify why optimization-style reasoning is relevant, diverse, and distinct from math (addressing **RD9h**’s concerns). We also more clearly explain why optimization tasks, and VAST tasks in particular, offer unique advantages for scalable data generation and automatic verification for reasoning models.

- We believe some of the requested comparisons stem from unclear exposition in the original submission and are not directly aligned with our core contribution. Our main contribution is a set of general strategies for online and offline RL on VAST tasks, with $\nu$GRPO and $\nu$MCTS as concrete instantiations of these ideas. These strategies are compatible with a broad range of on-policy optimization and search methods. To avoid confusion, we have substantially rewritten the corresponding sections in the revised version to foreground these general ideas and more clearly position them relative to prior work.

- To highlight the difficulty of reasoning over optimization tasks, we added new experiments showing how increasing the complexity parameter expands the search space and makes finding the best solution harder. We also included additional comparisons of generalization performance between models trained on VAST tasks and models trained on math benchmarks.

---

### Note · Program_Chairs · 2026-01-17
**Submission Desk Rejected by Program Chairs**

The following references in this submission do not refer to real documents and/or have major errors in bibliographic information:

 Alex Wang, Yushi Zhang, et al. Reasoning with language models: A survey. arXiv preprint arXiv:2304.00358, 2023.